# Resolving synaptic events using subsynaptically targeted GCaMP8 variants

**Jiawen Chen[1,2], Junhao Lin[1], Kaikai He[1,2], Luyi Wang[1], Yifu Han[1,2], Chengjie Qiu[1,2], Jasmine M Wheeler[3], Catherine M Daly[3], Gregory T Macleod[3], Dion K Dickman[1]\***

[1]University of Southern California, Department of Neurobiology, Los Angeles, United States; [2]USC Neuroscience Graduate Program, Los Angeles, United States; [3]Department of Physiology, Tulane University School of Medicine, New Orleans, United States

## eLife Assessment

In this **important** study, the authors engineered and characterised novel genetically encoded calcium indicators (GECIs) and an analytical tool (CaFire) capable of reporting and quantifying various sub-synaptic events, including miniature synaptic events, with a speed and sensitivity approaching that of intracellular electrophysiological recordings. They present **compelling** data validating this toolset, which will be of interest to neurobiologists studying synaptic calcium dynamics in various model systems.

**\*For correspondence:**
dickman@usc.edu

**Abstract** While genetically encoded $Ca^{2+}$ indicators are valuable for visualizing neural activity, their speed and sensitivity have had limited performance when compared to chemical dyes and electrophysiology, particularly at synaptic compartments. We addressed these limitations by engineering a suite of next-generation GCaMP8-based indicators, targeted to presynaptic boutons, active zones, and postsynaptic compartments at the *Drosophila* neuromuscular junction. We first validated these sensors to be superior to previous versions and synthetic dyes. Next, we developed a Python-based analysis program, *CaFire*, which enables the automated quantification of evoked and spontaneous $Ca^{2+}$ signals. Using *CaFire*, we show a ratiometric presynaptic GCaMP8m sensor accurately captures physiologically relevant presynaptic $Ca^{2+}$ changes with superior sensitivity and similar kinetics compared to chemical dyes. Moreover, we test the ability of an active zone-targeted, ratiometric GCaMP8m sensor to report differences in $Ca^{2+}$ between release sites. Finally, a newly engineered postsynaptic GCaMP8m, positioned near glutamate receptors, detects quantal events with temporal and signal resolution comparable to electrophysiological recordings. These next-generation indicators and analytical methods demonstrate that GCaMP8 sensors, targeted to synaptic compartments, can now achieve the speed and sensitivity necessary to resolve $Ca^{2+}$ dynamics at levels previously only attainable with chemical dyes or electrophysiology.

## Introduction

Evolution's ingenuity is evident in its exploitation of the calcium ion ($Ca^{2+}$) as a potent secondary messenger, essential for controlling a diverse array of physiological events in every cell. The basis for $Ca^{2+}$'s extraordinary signaling capacity lies in a formidable concentration gradient – maintained at nanomolar levels inside cells versus millimolar outside – that exceeds a thousand-fold. This carefully maintained difference allows for controlled $Ca^{2+}$ entry to rapidly amplify cytosolic $Ca^{2+}$ concentrations,

in turn activating specific Ca²⁺-sensitive proteins to initiate vital downstream signaling pathways (*Berridge et al., 2003*; *Clapham, 2007*). In neurons, this Ca²⁺-driven control system achieves a pinnacle of sophistication. The powerful Ca²⁺ gradient is harnessed to orchestrate crucial neuronal processes, from shaping synaptic strength at dendritic spines and modulating gene expression in response to activity, to the precise triggering of neurotransmitter release via synaptic vesicle (*Dittman and Ryan, 2019*; *Jackman and Regehr, 2017*; *McCarthy and Kavalali, 2024*). Therefore, the ability to directly observe and interpret these dynamic Ca²⁺ signals in neurons provides a crucial lens, offering profound insights into the complex cellular operations that define neuronal function and flow of information through neural circuits.

Given the paramount importance of Ca²⁺ signaling in virtually all aspects of neuronal function, an extensive series of genetically-encoded Ca²⁺ indicators (GECIs) have been engineered with progressively enhanced speed and sensitivity. The archetypal GECI, GCaMP, ingeniously utilized a modified green fluorescence protein (GFP) flanked by the Ca²⁺ binding protein Calmodulin (CaM) and a Ca²⁺/CaM-binding M13-like peptide (*Nakai et al., 2001*). Upon Ca²⁺ binding, a conformational change in the CaM-peptide interaction allosterically modulates GFP fluorescence. While a conceptual breakthrough, early GCaMP iterations suffered from relatively slow kinetics, limited dynamic range, and modest fluorescence changes (*Tian et al., 2009*). Consequently, a variety of sophisticated protein engineering strategies were employed to bolster GCaMP performance, leading to significant improvements represented in the GCaMP6 and GCaMP7 series, which offered faster kinetics and higher relative fluorescence changes (*Dana et al., 2019*; *Chen et al., 2013*). Finally, the subsequent development of the GCaMP8 series marked a substantial leap forward (*Zhang et al., 2023*). This enhanced performance was not merely incremental; it resulted from a deep understanding of the molecular mechanisms underlying GCaMP activation, involving meticulous optimization of the CaM/peptide interface for faster Ca²⁺ association and dissociation and other improvements that led to enhanced brightness and efficient conformational coupling (*Zhang and Looger, 2024*). These comprehensive engineering efforts yielded GCaMP8 variants that exhibit dramatically improved kinetics – both in rise and decay times – and sensitivity relative to previous generations. Indeed, their performance characteristics, including fluorescence changes, response speed, photostability, and linearity, became comparable or even superior to some of the best synthetic Ca²⁺ dyes, while retaining the crucial advantage of genetic targetability (*Zhang and Looger, 2024*). These innovations provide an opportunity to resolve Ca²⁺ dynamics at subcellular compartments with unprecedented resolution.

Many studies express GCaMP variants alone in the neuronal cytoplasm, a method that effectively captures the large, relatively slow Ca²⁺ fluxes in soma that correlate with action potential spiking (*Akerboom et al., 2012*; *Peron et al., 2015*; *Ziv et al., 2013*). However, Ca²⁺ transients within subcellular compartments, such as dendritic spines and presynaptic terminals, are characterized by significantly faster kinetics and occur within smaller, more restricted volumes than those in the general cytoplasm, posing distinct measurement challenges. Importantly, interpreting signals from GCaMPs expressed alone can be confounded by variations in sensor abundance due to uneven expression levels, differing protein stability, or photobleaching, any of which can obscure the accurate determination of true Ca²⁺ changes (*McMahon and Jackson, 2018*). Ratiometric indicators are, therefore, an attractive alternative, as they typically involve co-expression of Ca²⁺-sensitive and Ca²⁺-insensitive fluorescent proteins in a stoichiometric manner (*Zhang et al., 2021*). By normalizing the Ca²⁺-dependent signal to the Ca²⁺-independent one, ratiometric measurements can control for these variations in sensor concentration and provide a more reliable quantification of Ca²⁺ levels and dynamics. Hence, an optimal approach for accurately capturing fast and local subsynaptic Ca²⁺ fluxes would target an advanced, ratiometric GCaMP sensor, engineered for high sensitivity and rapid kinetics, to the particular compartment being investigated.

The *Drosophila* neuromuscular junction (NMJ) is a powerful and genetically tractable model system for dissecting Ca²⁺ signaling dynamics at a glutamatergic synapse. Its utility is magnified by the ability to combine sophisticated genetic manipulation with GECIs, advanced imaging, and established electrophysiological approaches to monitor synaptic events with remarkable spatiotemporal resolution (*Bellen et al., 2010*; *Frank et al., 2013*). Early investigations into Ca²⁺ dynamics at the fly NMJ employed chemical dyes (e.g. OGB-1, Fura-2), which, despite their rapid kinetics, present challenges in controlling loading concentrations at synaptic compartments (*Macleod et al., 2002*). The advent of GECIs offered an opportunity for targeted and consistent expression at subcellular

regions. Pioneering engineering of presynaptic GECIs enabled $Ca^{2+}$ visualization at axonal boutons or active zones (*Kiragasi et al., 2017*; *Cohn et al., 2015*; *Akbergenova et al., 2018*), but were often non-ratiometric for $Ca^{2+}$ sensing – complicating quantification – and employed older GCaMP variants with suboptimal kinetics and ΔF/F changes. Postsynaptic sensors evolved from using GCaMP2 to GCaMP6f, offering better sensitivity for detecting quantal events (*Newman et al., 2017*; *Peled et al., 2014*). More recently, GCaMP8f variants were employed (*Perry et al., 2022*; *Li et al., 2021*; *Han et al., 2023*), demonstrating promising sensitivity. However, other studies suggested that GCaMP8m may offer higher responses and similar kinetics, particularly in *Drosophila* (*Zhang et al., 2023*). These developments motivated us to engineer and validate optimal GECIs for the fly NMJ, aiming for performance that rivals chemical dyes for presynaptic imaging and electrophysiology for postsynaptic quantal event detection.

We have engineered and characterized a variety of next-generation GECI probes to resolve $Ca^{2+}$ dynamics at pre- and post-synaptic compartments at the *Drosophila* NMJ. While an expected trade-off between signal strength and kinetics was observed, GCaMP8m consistently delivered the highest responses with only modest reductions in speed relative to GCaMP8f. Notably, presynaptic GCaMP8m sensors captured physiologically important events with speed and sensitivity similar to or superior to that of chemical dyes, while postsynaptic GCaMP8m probes achieved quantal event detection with similar resolution to electrophysiological recordings. These features establish targeted ratiometric GCaMP8m sensors and CaFire analysis as complementary tools for resolving synaptic events using purely optical approaches.

## Results

### Engineering next-generation GCaMP sensors targeted to synaptic compartments

To engineer the next generation of synaptically targeted GCaMP sensors, we sought to improve upon previous sensors employed for $Ca^{2+}$ imaging at the *Drosophila* NMJ. In particular, we focused on three cassettes. First, to target GCaMP to presynaptic boutons, GCaMP6s was fused to the synaptic vesicle protein Synaptotagmin (SYT), generating UAS-SYT::GCaMP6s (*Cohn et al., 2015*), schematized in *Figure 1A*. More recently, this SYT::GCaMP fusion strategy was improved upon by replacing GCaMP6s with the faster and more sensitive GCaMP8f, and rendered ratiometric by fusion to the red-shifted, monomeric fluorescent protein mScarlet (*Bindels et al., 2017*), to generate UAS-SYT::mScarlet::GCaMP8f (Scar8f) (*Li et al., 2021*; *Figure 1A*). To potentially improve this indicator, we replaced GCaMP8f with GCaMP8m, which is reported to exhibit higher sensitivity with only minor reductions in kinetics (*Zhang et al., 2023*), and replaced mScartlet1 with mScarlet3, which encodes improvements in brightness and photophysical characteristics (*Gadella et al., 2023*), to make the ratiometric GECI Scar8m (*Figure 1A*). Expression of each sensor with the motor neuron driver OK319-Gal4 and immunostaining demonstrated the expected trafficking to presynaptic boutons and co-localization with SYT (*Figure 1B*).

Next, to target GCaMP to the sites of presynaptic $Ca^{2+}$ influx, active zones (AZs), we improved upon a previous indicator, UAS-BRPshort::mCherry::GCaMP6s (*Kiragasi et al., 2017*). This strategy used a 'short' fragment of the AZ scaffold Bruchpilot (BRP) (*Schmid et al., 2008*), which traffics to AZs, fused to mCherry and GCaMP6s for ratiometric $Ca^{2+}$ imaging at AZs (*Figure 1C*). To improve upon this design, we replaced mCherry with the improved mScarlet and GCaMP6s with GCaMP8f to make UAS-BRPshort::mScarlet::GCaMP8f (Bar8f) or GCaMP8m and mScarlet3 to make Bar8m (*Figure 1C*). Expression of these sensors in motor neurons followed by immunostaining also confirmed the expected trafficking to AZs and co-localization with BRP (*Figure 1D*). Finally, we worked to improve upon a postsynaptic $Ca^{2+}$ sensor. A previous cassette called SynapGCaMP6f fused GCaMP6f to a Shaker PDZ motif to target the sensor near postsynaptic NMJ glutamate receptors (GluRs) under the control of a muscle-specific enhancer to enable 'quantal imaging' of single synaptic vesicle release events (*Newman et al., 2017*; *Figure 1E*). A recent improvement replaced GCaMP6f with GCaMP8f (*Han et al., 2022*), and here we engineered GCaMP8m to make SynapGCaMP8m (*Figure 1E*). Immunostaining of NMJs expressing each sensor confirmed proper trafficking and localization to postsynaptic densities near GluRs (*Figure 1F*). In addition, GCaMP is a $Ca^{2+}$ buffer and may, therefore, impact synaptic development and/or function (*McMahon and Jackson, 2018*). Thus, we recorded from NMJs

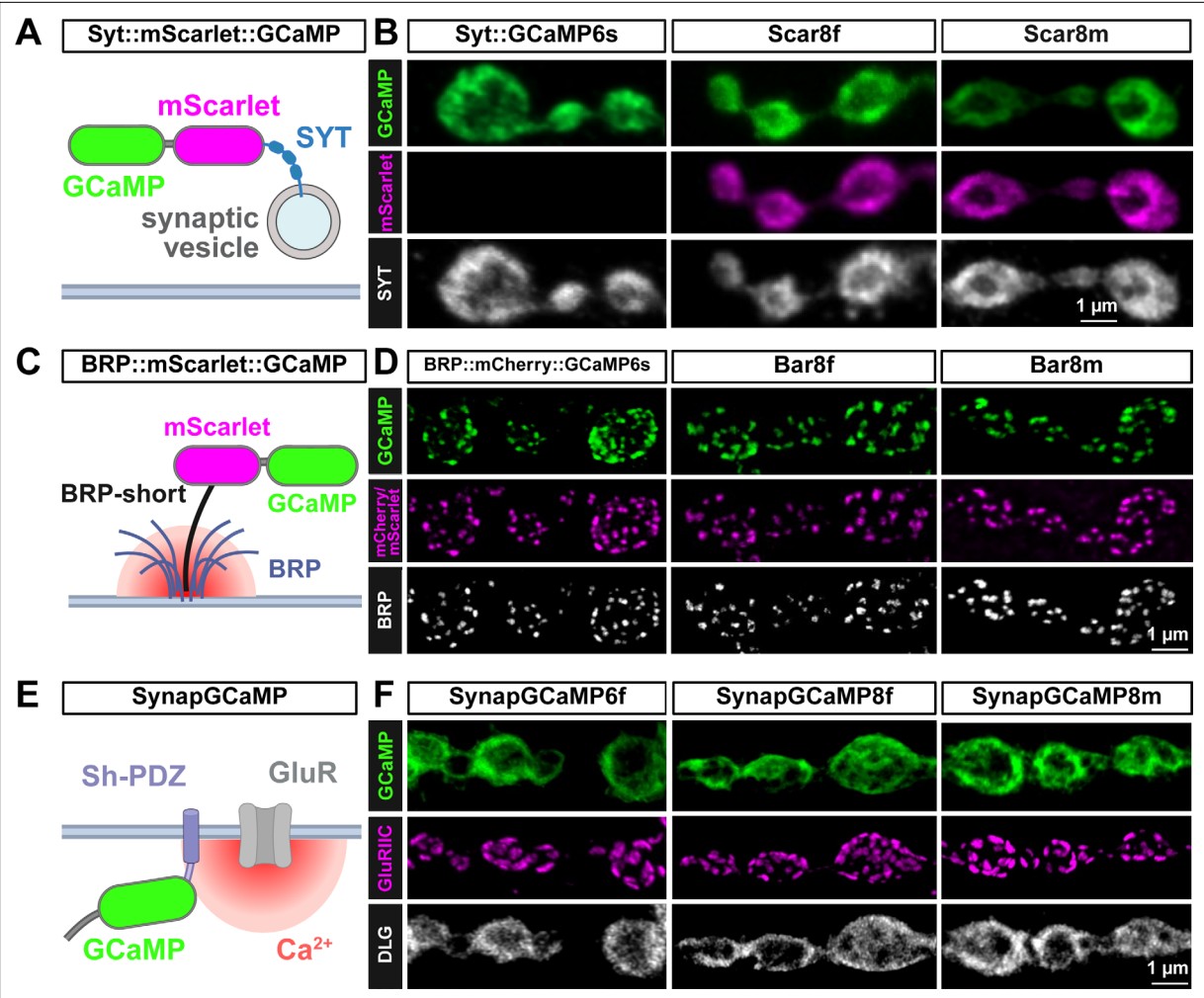

**Figure 1.** GCaMP indicators targeted to pre- and post-synaptic compartments. (**A**) Schematic of the presynaptic ratiometric Syt::mScarlet::GCaMP (Scar8f/Scar8m) Ca²⁺ indicators showing localization to synaptic vesicles via fusion to the Ca²⁺ sensor Synaptotagmin (SYT). (**B**) Representative images of neuromuscular junctions (NMJs) expressing the indicated reporter driven in motor neurons with the OK319-GAL4 driver (*w1118; OK319-GAL4/UAS-Scar8f*) immunostained with anti-GFP (GCaMP) and anti-Synaptotagmin (SYT). Note that endogenous mScarlet signals were obtained without antibody labeling. (**C**) Schematic of the BRP::mScarlet::GCaMP8f/8 m (Bar8f/Bar8m) ratiometric Ca²⁺ indicator, which targets GCaMP to active zones via fusion to the Bruchpilot (BRP)-short protein (*Schmid et al., 2008*). (**D**) Representative images of NMJs expressing the indicated reporter driven in motor neurons (*w;OK6-GAL4/Bar8f* and *w;OK6-GAL4/Bar8m*) immunostained with anti-GFP (GCaMP) and anti-BRP. Note that native mCherry or mScarlet signals were obtained without antibody labeling. (**E**) Schematic of the SynapGCaMP indicator, which targets GCaMP to postsynaptic compartments via a Shaker PDZ domain.(*Newman et al., 2017*) (**F**) Representative images of NMJs expressing the indicated reporter (*w;MHC-CD8-GCaMP6f-Sh/+;+, w;;MHC-CD8-GCaMP8f-Sh/+, w;;MHC-CD8-GCaMP8m-Sh/+*) immunostained with anti-GFP (GCaMP), -GluRIIC (glutamate receptors), and -DLG (postsynaptic density). All confocal images in this figure were deconvolved in Huygens (CMLE) prior to display (see Methods). Created with BioRender.com.

The online version of this article includes the following figure supplement(s) for figure 1:

**Figure supplement 1.** Genetically-encoded Ca2+ indicator (GECI) expression does not perturb synaptic transmission at the *Drosophila* neuromuscular junction (NMJ).

expressing the indicators introduced above and found that expression of the GCaMP sensors did not perturb synaptic transmission, with the exception of Scar8f, which showed a moderate reduction in EPSP amplitudes (*Figure 1—figure supplement 1*). Collectively, these tools established the foundation of our study to go on to benchmark each sensor and determine the optimal indicator and their speed and sensitivity to resolve pre- and post-synaptic Ca²⁺ events and dynamics.

## Automated analysis of Ca²⁺ imaging data

Current approaches to analyze synaptic $Ca^{2+}$ imaging data either repurpose software designed to analyze electrophysiological data or use custom software developed by groups for their own specific needs, including measuring spike frequency across many neurons (*Müller and Davis, 2012*; *Giovannucci et al., 2019*; *Blum et al., 2021*; *Xing and Wu, 2018*). We, therefore, developed a new Python-based software platform capable of automated detection and quantification of synaptic $Ca^{2+}$ events. This program, which we named 'CaFire,' analyzes imaging data followed by optional processing, such as deconvolution (*Figure 2A*). The $Ca^{2+}$ imaging data are then imported into ImageJ to select Regions of Interest (ROIs), which are finally exported to a graphical program plotting event time and intensity data (e.g. Microsoft Excel; see methods for details). The CaFire software reads the Excel file and performs all downstream analysis.

CaFire can analyze both presynaptic evoked and postsynaptic quantal $Ca^{2+}$ events with a user-friendly graphical interface that enables parameter adjustment and event visualization (*Figure 2B*). For both quantal and evoked events, peak detection is based on customizable parameters, including threshold amplitude, peak width, and minimum inter-peak distance. These parameters can be adjusted to optimize detection sensitivity and specificity, while misidentified events can be manually corrected. Once events are identified, CaFire then quantifies key parameters, such as peak amplitude ($\Delta F/F$), rise time constants based on an exponential growth formula, and decay time constants using a natural logarithmic decay formula (*Figure 2C*; see methods). In addition, evoked $Ca^{2+}$ events can be partitioned at different stimulation frequencies (e.g. 1 Hz, 5 Hz) to assess frequency-dependent dynamics. All quantification results are displayed in real-time and can be exported in a tabular format for downstream statistical analysis. This automated workflow significantly reduces analysis time and user variability in $Ca^{2+}$ imaging data analysis when compared to a variety of other platforms we have tried, providing uniform, accurate, and reproducible measurements. CaFire's ability to analyze both evoked and quantal events makes it a versatile tool for quantifying synaptic $Ca^{2+}$ imaging data.

## Benchmarking presynaptic Ca²⁺ sensors

We first sought to directly compare the performance of Scar8f with chemical $Ca^{2+}$ indicator Oregon Green BAPTA-1 (OGB-1) under the same conditions. Images of the type-Ib terminal on muscle #6 (MN6/7-Ib) terminals were captured at 303 frames per second on a confocal microscope (*Figure 3A*), evoking fluorescence transients at 1 Hz. In all aspects, Scar8f matched or exceeded the performance of OGB-1 (*Figure 3B–E*). Specifically, GCaMP8f displayed higher sensitivity compared to OGB-1 and exhibited a faster rate of decay (*Figure 3B and C*). The terminals were also challenged with a 21.3 Hz train of stimuli, where individual $Ca^{2+}$ transients were easily discerned in both GCaMP8f and OGB-1 traces (*Figure 3D*). However, GCaMP8f fluorescence summated to a far greater degree, representing greater frequency facilitation of the $Ca^{2+}$ transients (*Figure 3D and E*). Finally, we compared the evoked $Ca^{2+}$ signals from cytosolic GCaMP8m (RSET-8m, *Reagents et al., 2024*) to Scar8m (*Figure 3F*). We observed an almost twofold increase in the amplitude of the Scar8m response in Scar8m compared to cytosolic GCaMP8m (*Figure 3G and H*). Thus, Scar8f/m, targeted to synaptic vesicles, exhibits superior sensitivity and kinetics compared to the chemical dye OGB-1 and cytosolic GCaMP8m indicators.

Next, we systematically evaluated the performance of the presynaptically targeted $Ca^{2+}$ indicators SYT::GCaMP6s, Scar8f, and Scar8m at MN-Ib terminals. Live confocal imaging confirmed robust expression of all sensors at presynaptic boutons, with colocalized GCaMP and mScarlet fluorescence (*Figure 4A and B*). $Ca^{2+}$ transients were elicited by single action potential (AP) electrical stimulation, and fluorescence responses were recorded and analyzed (*Figure 4C and D*). Scar8m produced the highest amplitudes of GCaMP/mScarlet ($\Delta R/R$) ratios, with GCaMP8m showing 345.7% higher $\Delta F/F$ over GCaMP6s, and 55.7% higher compared to GCaMP8f (*Figure 4D and E*). Both Scar8f and Scar8m clearly outperformed GCaMP6s in both sensitivity and speed (*Figure 4D and E*), as expected. Importantly, while GCaMP8f exhibited slightly faster decay time constants (66.6 ms vs 99.2 ms), rise time constants were not significantly different between the two sensors (5.3 ms vs 7.0 ms) (*Figure 4E*). Thus, Scar8m exhibited an optimal balance between kinetics and signal strength, depending on the objectives of the experimental approach.

To assess sensor performance during repetitive activity, we tested responses to five stimuli at 5 Hz and 10 Hz stimulation trains. While responses were muted using SYT::GCaMP6s, Scar8f, and Scar8m consistently demonstrated robust responses and separation across repetitive stimulation

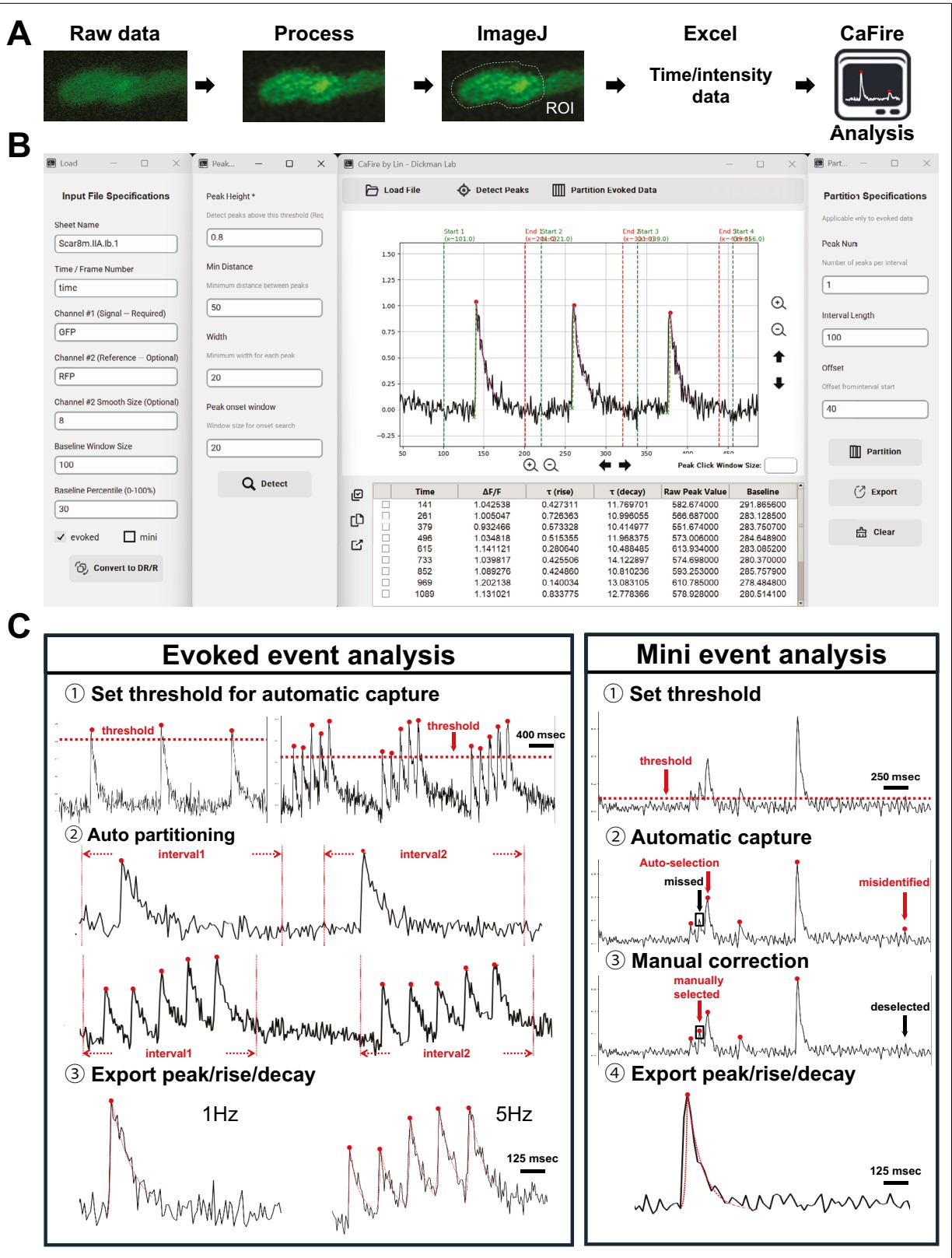

**Figure 2.** 'CaFire' - a Python-based analysis program for quantifying synaptic Ca$2^{2+}$ imaging data. (**A**) Workflow showing how Ca$^{2+}$ imaging data and downstream analysis is performed. Raw time-lapse movies are processed with SVI Huygens software to correct and deconvolve image artifacts. Regions of interests (ROIs) are then selected in ImageJ Fiji software and intensities are extracted. Data are exported to Excel for manual inspection or directly analyzed using CaFire software. (**B**) Screenshots of the CaFire user interface. The left panels allow users to specify input file properties and analysis

*Figure 2 continued*

settings, and the second panel from the left shows the peak-detection criteria. The central panel displays fluorescence traces with automatically detected Ca²⁺ events overlaid and the event-based partition results. Detected events and their extracted parameters are listed in the data table below. The right panel shows the partition specifications used to segment evoked responses. (**C**) Examples of two distinct analysis pipelines implemented in CaFire. *Evoked event analysis*: (1) Thresholds are set for automatic peak detection; (2) Events are automatically partitioned based on stimulus intervals (e.g. 1 Hz and 5 Hz); (3) Parameters, such as peak amplitude, rise time constant ($\tau_{rise}$), and decay time constant ($\tau_{decay}$) are calculated using exponential fits. *Mini event analysis*: (1) Users define amplitude thresholds for event detection; (2) CaFire automatically identifies candidate events; (3) Missed or misidentified events can be manually corrected; (4) Event parameters are exported for each validated event.

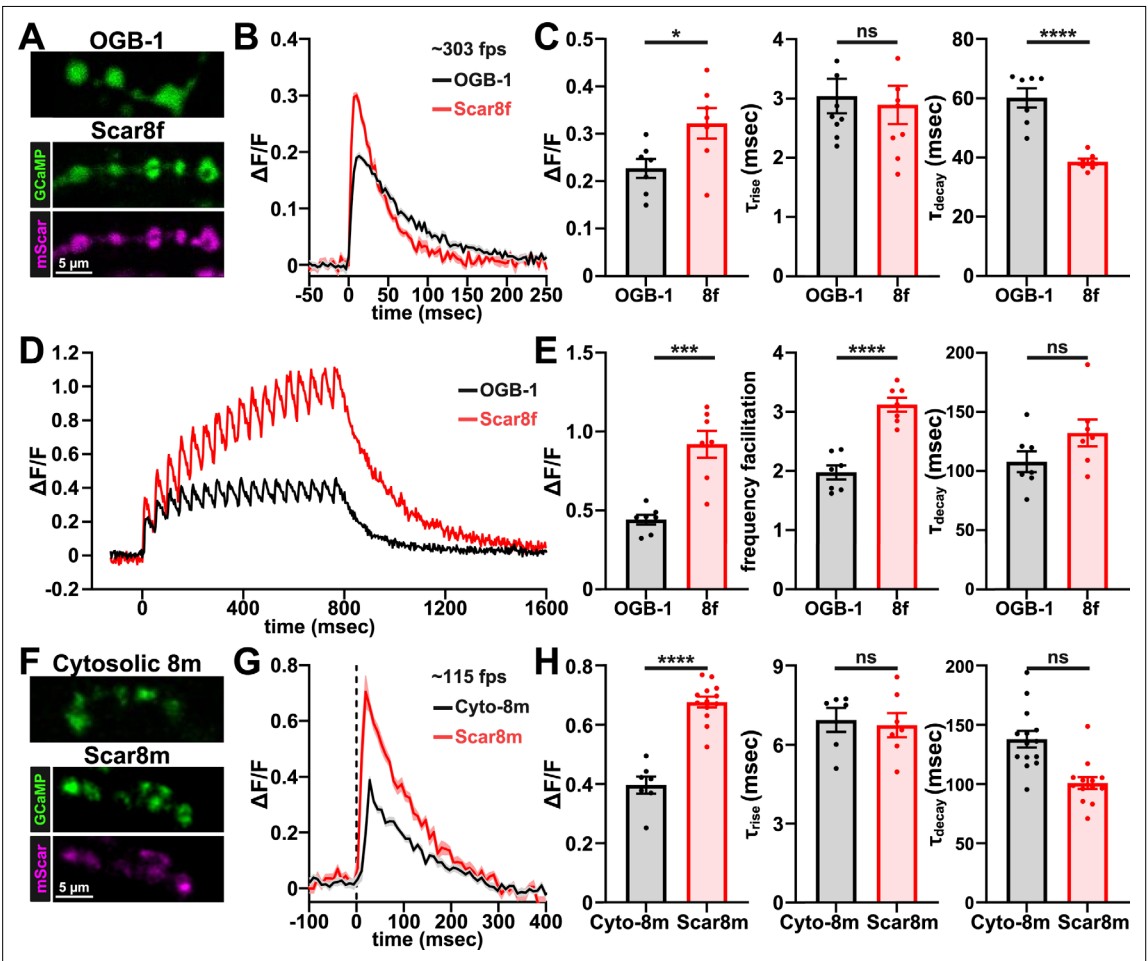

**Figure 3.** Benchmarking presynaptic GCaMP8 to synthetic dyes and cytosolic sensors. (**A**) Confocal images showing presynaptic loading of the synthetic dye OGB-1 and presynaptic expression of Scar8f at MN-Ib boutons. (**B**) Example single-action potential (AP) evoked presynaptic Ca²⁺ transients reported by OGB-1 (black) and GCaMP8f (red). Traces represent the average of 30 APs from a single preparation acquired at 303 fps. (**C**) Quantification of response amplitude (ΔF/F), rise time, and decay times from single-AP Ca²⁺ transients reported by OGB-1 and Scar8f. (**D**) Example train-evoked Ca²⁺ transients reported by OGB-1 (black) and Scar8f (red) during three trains of 10 APs delivered at 21.3 Hz. (**E**) Quantification of train-evoked Ca²⁺ signals showing amplitude (ΔF/F) of the final Ca²⁺ transient in the train (17th), measured from baseline; frequency facilitation, calculated as the amplitude of the 17th transient divided by the amplitude of the first; and decay time constant ($\tau_{decay}$) measured after the final transient in the train. (**F**) Confocal neuromuscular junctions (NMJs) images showing presynaptic expression of the cytosolic sensor RSET-8m (Cytosolic GCaMP8m) and Scar8m. (**G**) Example single-AP evoked Ca²⁺ transients reported by RSET-8m (black) and Scar8m (red) acquired at ~115 fps. (**H**) Quantification of amplitude (ΔF/F), rise time ($\tau_{rise}$), and decay time constants ($\tau_{decay}$) from single-AP Ca²⁺ transients reported by RSET-8m (*w;OK319-GAL4/+; UAS-RSET-GCaMP8m/+*) and Scar8m (*w;OK319-GAL4/+; UAS-Syt::mScarlet::GCaMP8m/+*). Shaded traces and bars show mean ± SEM. Statistical comparisons are unpaired two-tailed t-tests; significance is indicated as *$p<0.05$, **$p<0.01$, ***$p<0.001$, ****$p<0.0001$; ns, not significant. Detailed statistics, including *p*-values are provided in ***Supplementary file 1***. Created with BioRender.com.

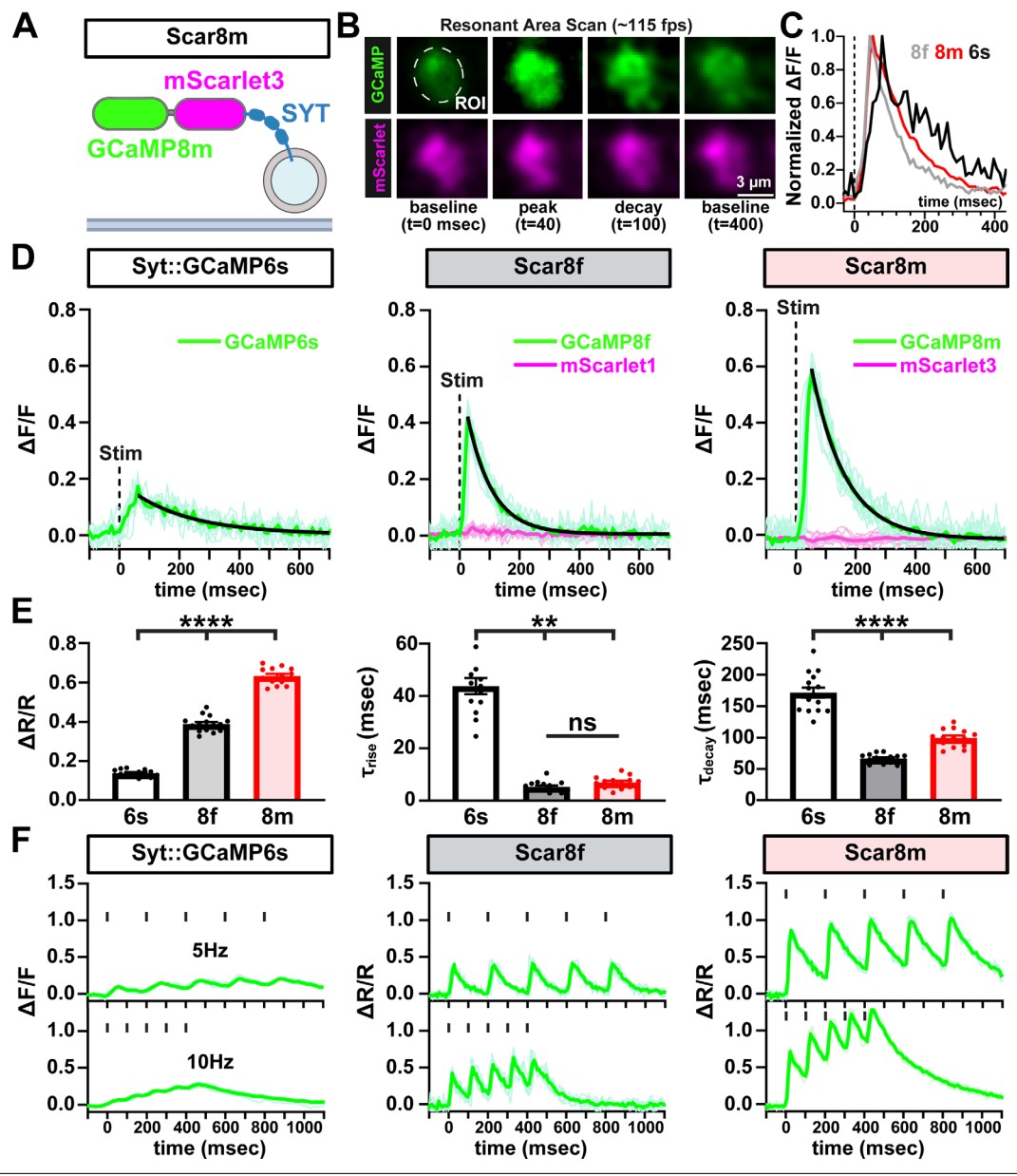

**Figure 4.** Scar8m is an optimal presynaptic Ca$^{2+}$ indicator. (**A**) Schematic of the UAS-Syt::mScarlet3::GCaMP8m (Scar8m) ratiometric Ca$^{2+}$ indicator consisting of mScarlet3 and GCaMP8m targeted to synaptic vesicles via fusion to Synaptotagmin (SYT). (**B**) Representative images of a MN-Ib bouton expressing Scar8m (*w;OK319-GAL4/+;Scar8m/+*) resonant scanned at ~115 fps. Fluorescence from GCaMP8m (green) and mScarlet3 (magenta) is shown at baseline, peak, decay, and recovery to baseline. (**C**) Normalized Ca$^{2+}$ signals following single action potential (AP) stimulation. Averaged traces compare the kinetics of GCaMP6s, GCaMP8f, and GCaMP8m. (**D**) Representative GCaMP and mScarlet signals recorded from Syt::GCaMP6s (GCaMP only), Scar8f, and Scar8m (GCaMP plus mScarlet) in response to single AP stimuli. Thin traces are sequential single-trial sweeps from the same neuromuscular junction (NMJ); the thick colored trace is the mean trace after stimulus alignment. The black curve indicates a single-exponential decay fit used to estimate $\tau$. (**E**) Quantification of average peak amplitude (ΔR/R, GCaMP/mScarlet ratios), rise time constant ($\tau_{rise}$), and decay time constant ($\tau_{decay}$) from the indicated sensors. Scar8m yields significantly higher peak ΔR/R signals, similar rise time kinetics, and a modestly slower decay compared to Scar8f. (**F**) Presynaptic Ca$^{2+}$ responses of the indicated sensors to 5 Hz and 10 Hz stimulation trains. Vertical black ticks above the traces indicate the timing of stimulation pulses. Error bars represent ± SEM. Significance is indicated as *$p<0.05$, **$p<0.01$, ***$p<0.001$, ****$p<0.0001$; ns, not significant; detailed statistics, including *p*-values are presented in ***Supplementary file 1***. Created with BioRender.com.

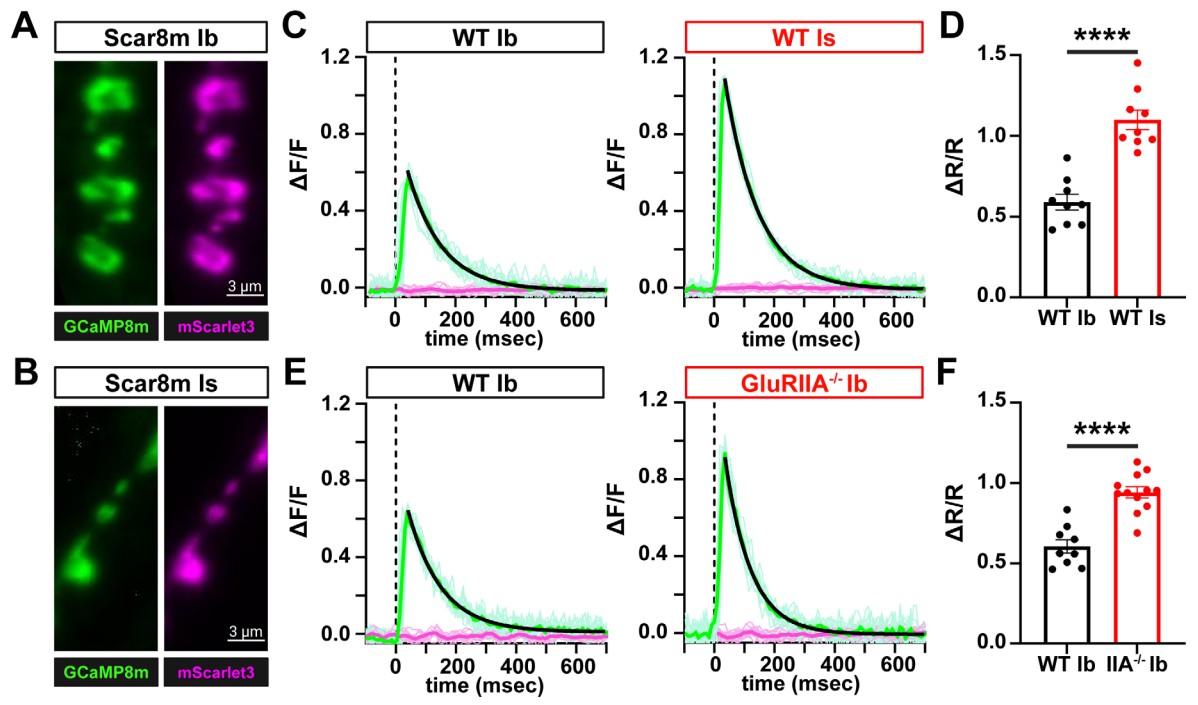

**Figure 5.** Scar8m captures differences in Ca2²⁺ levels between motor neuron subtypes and after plasticity. (**A, B**) Representative images of Scar8m expressed at both MN-Ib (**A**) and MN-Is (**B**) motor neuron subtypes immunostained with anti-GFP. (**C**) ΔF/F traces of GCaMP8m and mScarlet3 responses from single action potential (AP) stimulation at MN-Ib and MN-Is, with ~2 x higher responses observed at MN-Is over -Ib, as expected. Thin traces are sequential single-trial sweeps from the same neuromuscular junction (NMJ); the thick colored trace is the mean trace after stimulus alignment. The black curve indicates a single-exponential decay fit used to estimate $\tau$. (**D**) Quantification of ΔR/R responses from the two inputs. (**E**) ΔF/F traces of GCaMP8m and mScarlet3 responses from single AP stimulation at MN-Ib in wild-type (*w;OK319-GAL4/+;Scar8m/+*) and *GluRIIA* mutants (*w;OK319-GAL4,GluRIIA^PV3^/GluRIIA^PV3^;Scar8m/+*), which express presynaptic homeostatic plasticity (PHP). Note the enhanced presynaptic Ca²⁺ levels induced after PHP plasticity. (**F**) Quantification of ΔR/R responses from the indicated genotypes. Error bars represent ± SEM. Significance is indicated as *$p<0.05$, **$p<0.01$, ***$p<0.001$, ****$p<0.0001$; ns, not significant; detailed statistics, including *p*-values are presented in **Supplementary file 1**. Created with BioRender.com.

(*Figure 4F*). Together, we conclude that because of the superior sensitivity, ratiometric properties, and fast kinetics, Scar8m is an optimal GECI to quantify stimulated Ca²⁺ responses from presynaptic terminals.

To determine whether the sensitivity of Scar8m is sufficient to quantitatively report biologically relevant differences in evoked presynaptic Ca²⁺ responses, we compared evoked responses between MN-Ib and MN-Is terminals. Previous studies using chemical indicators have shown MN-Is exhibits ~2–3 x increased Ca²⁺ responses over MN-Ib (*Lu et al., 2016a, He et al., 2023*). First, we confirmed robust Scar8m expression in both MN-Ib and -Is using the OK319-Gal4 driver (*Figure 5A and B*). Next, Scar8m responses to single AP stimulation revealed a >two fold increase at MN-Is terminals compared to MN-Ib (*Figure 5C and D*). Next, we assessed changes in presynaptic Ca²⁺ responses at MN-Ib in wild-type and *GluRIIA* mutants, in which a process called presynaptic homeostatic potentiation (PHP) is known to be induced (*He and Dickman, 2025*; *Davis and Müller, 2015*). In PHP, diminished postsynaptic GluR functionality is offset through a homeostatic signaling system which enhances neurotransmitter release (*Goel and Dickman, 2021*; *Delvendahl and Müller, 2019*). PHP induces an increase in presynaptic Ca²⁺ influx to promote additional synaptic vesicle release (*Müller and Davis, 2012*; *Chien et al., 2025*), where chemical indicators, such as OGB-1 have shown ~30% increase in Ca²⁺ levels after PHP (*Müller and Davis, 2012*). Importantly, Scar8m responses in *GluRIIA* mutant MN-Ib boutons showed significantly elevated ΔR/R amplitudes compared to wild-type, with a ~53% enhancement (*Figure 5E and F*). Thus, Scar8m can resolve biologically relevant differences in presynaptic Ca²⁺ with high sensitivity, on par with or exceeding what chemical indicators have shown.

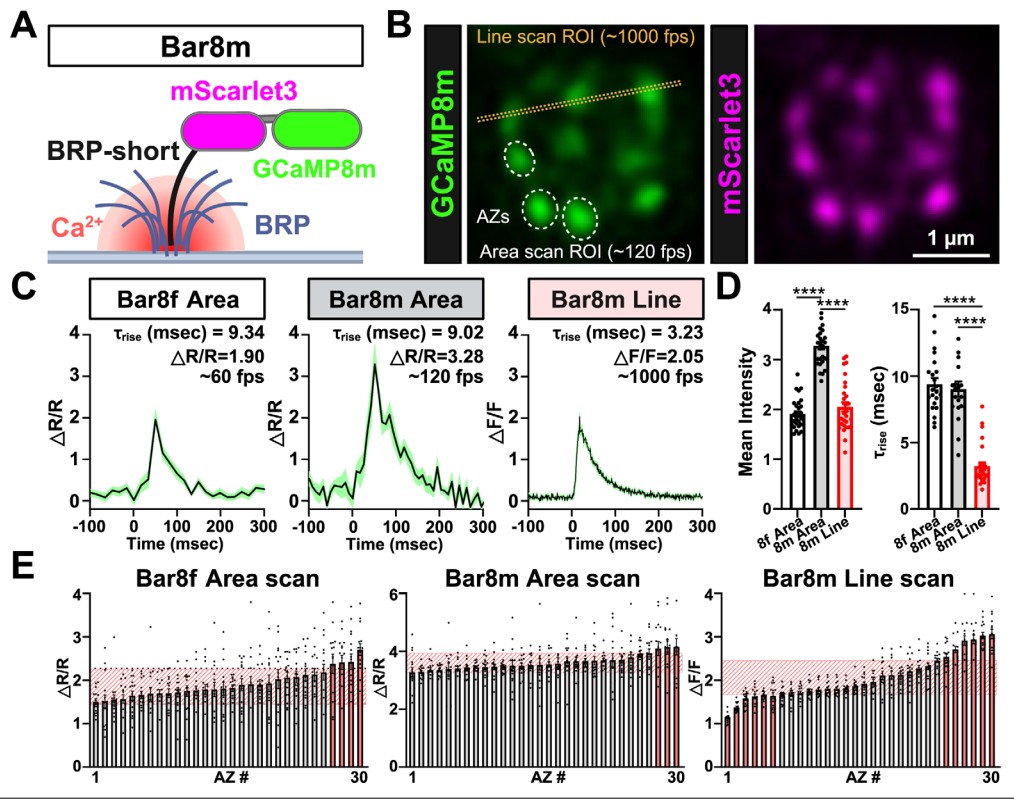

**Figure 6.** Evaluating the ability of Bar8f and Bar8m to capture active zone–specific Ca²⁺ changes. (**A**) Schematic of the BRP::mScarlet3::GCaMP8m (Bar8m) ratiometric Ca²⁺ indicator, consisting of GCaMP8m and mScarlet3 fused to the Bruchpilot (BRP)-short domain, which traffics to individual active zones (AZs). (**B**) Live confocal image of a single MN-Ib bouton expressing Bar8m. Dashed circles indicate active zones used for area-scan measurements, and the yellow dotted line indicates the line-scan regions of interest (ROI) used for high-speed imaging. (**C**) Example Ca²⁺ transients from individual AZs recorded using Bar8f resonant area scans (~60 fps; left), Bar8m resonant area scans (~120 fps; middle), and Bar8m line scans (~1000 fps; right). Black traces show the mean ratiometric response (ΔR/R) for Bar8f and Bar8m area scans and the mean ΔF/F response for Bar8m line scans from a representative neuromuscular junction (NMJ); green shading denotes ± SEM across repeated trials. (**D**) Quantification of mean evoked response amplitude and $\tau_{rise}$ for the three imaging conditions shown in (**C**). (**E**) Bar graph showing peak ΔR/R (or ΔF/F) values from individual AZs collected across multiple boutons from different NMJs for Bar8f area scans, Bar8m area scans, and Bar8m line scans. Each bar represents a single active zone (sorted by amplitude within each condition). Light red shaded boxes indicate the interquartile range (IQR; Q1–Q3) of individual data points for each active zone, and active zones whose mean responses fall outside this interquartile range (IQR) range are highlighted in pink. Significance is indicated as *$p<0.05$, **$p<0.01$, ***$p<0.001$, ****$p<0.0001$; ns, not significant; detailed statistics, including $p$-values are presented in ***Supplementary file 1***. Created with BioRender. com.

The online version of this article includes the following figure supplement(s) for figure 6:

**Figure supplement 1.** Additional analysis of calcium signal amplitudes versus AZ size for Bar8f.

## Evaluating the Bar8m sensor to resolve local active zone Ca²⁺ signals

Differences in Ca²⁺ influx and buffering at individual release sites contribute to functional differences at synapses (*Dittman and Ryan, 2019*; *Jackman and Regehr, 2017*; *McCarthy and Kavalali, 2024*). While Scar8m is an excellent sensor to define spatially-averaged presynaptic Ca²⁺ levels at individual boutons, the ability to resolve the heterogeneous Ca²⁺ signals within individual boutons during neuro-transmission would be a powerful tool to interrogate presynaptic function. Thus, we engineered the Bar8m sensor to localize a ratiometric GCaMP8m/mScarlet3 cassette to AZs by fusion to the BRP-short motif (*Figure 6A*). While some previous studies have suggested a similar approach can resolve AZ-specific differences in Ca²⁺ influx (*Akbergenova et al., 2018*), from the outset, we were cognizant of many challenges to this line of investigation. First, theoretical calculations suggest that Ca²⁺ diffuses

and equilibrates at a rate of <3 msec across an entire bouton, faster than most sensors would respond before receiving $Ca^{2+}$ contributions from other release sites (*Maravall et al., 2000*; *Sinha et al., 1997*; *Helmchen et al., 1997*). Second, it is difficult to resolve individual AZs from live imaging of boutons, as many are closely spaced when flattened 2D images are taken. Finally, at the *Drosophila* NMJ, individual AZs labeled by BRP are very small, with diameters averaging ~300 nm (*Ramesh et al., 2021*). To spatially resolve these discrete AZs during $Ca^{2+}$ imaging, high spatial resolution is required, which often necessitates reducing the scanning rate. Furthermore, when employing dual-channel imaging to simultaneously capture signals from GCaMP and mScarlet, sequential scanning is typically implemented to minimize spectral crosstalk between channels. These technical constraints collectively limit the achievable frame rate for AZ-level $Ca^{2+}$ imaging to ~120 frames per second (fps) using resonant area scanning (*Chen et al., 2024*), although faster frame rates can be achieved with line scanning or CMOS cameras.

Despite these acknowledged challenges and potential confounds, we sought to test the ability of Bar8m to resolve local AZ $Ca^{2+}$ dynamics following single AP stimulation. Individual AZs were identified at MN-Ib boutons using basal mScarlet3 fluorescence, then evoked $Ca^{2+}$ signals were measured using either resonant area or line scanning (*Figure 6B*). Peak ΔR/R values were measured at individual AZs (*Figure 6B and C*), where typical ΔR/R values at individual AZs ranged from ~1.5–2.5 (Bar8f), 2.0–3.5 (Bar8m) using resonant area or line scanning (*Figure 6D*). These results, while taking into account the caveats detailed above, are consistent with the theoretical estimates of $Ca^{2+}$ levels being equilibrated across all AZs before the $Ca^{2+}$ indicator can capture local differences, at least using relatively slow resonant area scanning approaches.

To further assess $Ca^{2+}$ levels at individual AZ captured by the Bar8f/m sensor, we analyzed our imaging data in more detail. We plotted individual AZ ΔR/R values from lowest to highest using resonant area scanning (Bar8f and Bar8m), or line scanning (Bar8m) (*Figure 6E*, *Figure 6—figure supplement 1*). While it was clear that the relatively slow area scanning was likely insufficient to resolve AZ-specific differences in $Ca^{2+}$ levels, with only 3-4/30 AZs showing significant differences, fast line scanning of Bar8m was capable of capturing significant differences in 12/40 AZs in our imaging system. While we interpret the observed relative consistency of ΔR/R at individual AZs using resonant area scanning to be unable to reliably resolve local $Ca^{2+}$ changes before equilibration within an individual bouton, faster approaches (line scanning or sCMOS cameras) might be able to detect such heterogeneity.

## SynapGCaMP8 sensors resolve quantal events at postsynaptic compartments

Finally, we evaluated the performance of SynapGCaMP6f, SynapGCaMP8f, and SynapGCaMP8m sensors to monitor postsynaptic 'quantal' events at the *Drosophila* NMJ. These quantal events reflect the rapid ionic influxes that result from the spontaneous release of single synaptic vesicles, which open postsynaptic GluRs and allow passage of $Na^+$ and $Ca^{2+}$ ions (*Grienberger and Konnerth, 2012*). Electrophysiological methods are the gold standard to report these mEPSP events, and we were particularly interested in determining whether SynapGCaMP variants are capable of detecting quantal events with similar sensitivity.

We first confirmed that all SynapGCaMP variants expressed well and trafficked to postsynaptic compartments, localizing with the scaffold DLG while encompassing but being distinct from glutamate receptive fields, which is particularly apparent with super-resolution STED microscopy (*Figure 7A*). Using resonant area scanning at ~115 fps, we imaged and analyzed single quantal $Ca^{2+}$ events for each SynapGCaMP variant. Averaged traces revealed a progressive improvement in response kinetics and sensitivity across the indicator series (*Figure 7B and C*). SynapGCaMP6f exhibited modest sensitivity, with ΔF/F values of 0.27 and relatively slow kinetics ($\tau_{rise}$=21 ms, $\tau_{decay}$=99 ms), while SynapGCaMP8f achieved faster responses and higher sensitivity (ΔF/F=0.35; $\tau_{rise}$=14 ms, $\tau_{decay}$=42 ms). SynapGCaMP8m exhibited the highest peak amplitude (ΔF/F=0.58), with similarly fast rise times ($\tau_{rise}$=14 ms) and moderate slowing of the decay ($\tau_{decay}$=67 ms), striking an optimal balance between speed and sensitivity (*Figure 7C and E*). SynapGCaMP quantal signals appeared to qualitatively reflect the same events measured with electrophysiological recordings (*Figure 7D*). We conclude that SynapGCaMP8m is an optimal indicator to measure quantal transmission events at the synapse.

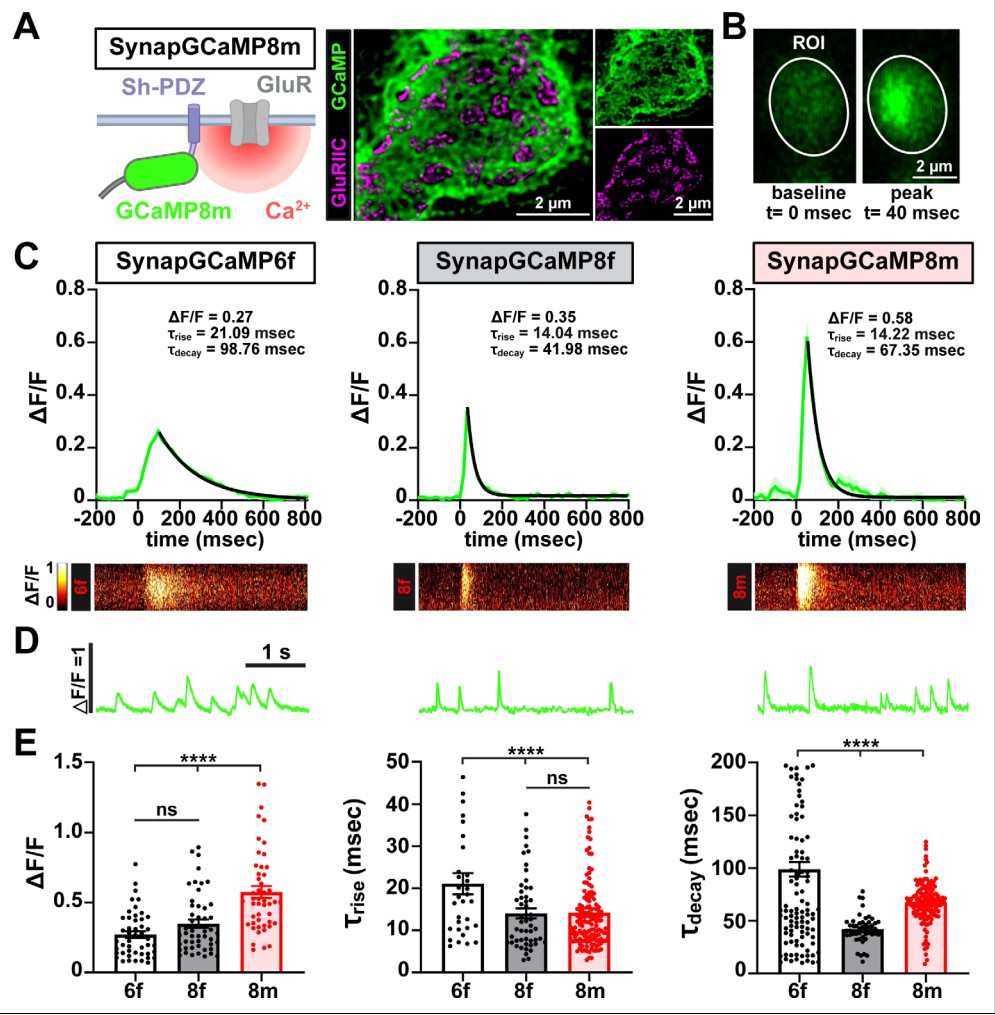

**Figure 7.** SynapGCaMP8m is an optimal postsynaptic Ca²⁺ indicator. (**A**) Schematic of the SynapGCaMP8m reporter, with GCaMP targeted to postsynaptic compartments near glutamate receptors via a Shaker-PDZ motif. Super-resolution image using STED microscopy showing the GCaMP8m reporter is localized outside of GluRs. (**B**) Live confocal images of muscle 6 neuromuscular junction (NMJ) boutons expressing SynapGCaMP8m were performed using resonant area scans. The indicated regions of interest (ROI) shows representative frames at baseline and peak quantal Ca²⁺ transients acquired at ~115 fps. (**C**) Averaged single miniature Ca²⁺ events recorded from SynapGCaMP6f, –8 f, and –8 m. The thick green trace shows the mean ΔF/F waveform and the light green shading indicates ± SEM; the black curve over the decay phase is a single-exponential fit used to estimate $\tau_{decay}$. SynapGCaMP8m yields the highest peak signal and maintains rapid kinetics. The corresponding heatmaps below were generated from a single vertical line scan extracted from the representative miniature-event ROI and visualize a spatiotemporal fluorescence dynamics (ΔF/F) along that line over time. (**D**) Representative Ca²⁺ events reporting individual miniature transmission in the indicated sensors. (**E**) Quantification of ΔF/F peak amplitude, and rise, and decay time constants ($\tau_{rise}$ and $\tau_{decay}$) for each SynapGCaMP variant. All comparisons in bar graphs are statistically significant unless 'ns' is shown. Error bars indicate ± SEM. Significance is indicated as *$p<0.05$, **$p<0.01$, ***$p<0.001$, ****$p<0.0001$; ns, not significant; detailed statistics, including $p$-values are presented in **Supplementary file 1**. Created with BioRender.com.

Next, we systematically assessed the ability of the SynapGCaMP variants to resolve quantal events. In particular, we switched to a widefield Ca²⁺ imaging system mounted on an electrophysiology rig so that we could perform simultaneous Ca²⁺ imaging and electrophysiological recordings of mEPSP events. To simplify our analysis, we isolated miniature events from MN-Ib inputs by silencing MN-Is transmission using selective expression of BoNT-C (**Han et al., 2022**) (see Methods). Sharp electrode intracellular recordings and Ca²⁺ imaging from muscle 6 MN-Ib boutons expressing SynapGCaMP6f, –8 f, or –8 m revealed increasing fidelity of quantal events (**Figure 8A and B**). Synchronized GCaMP

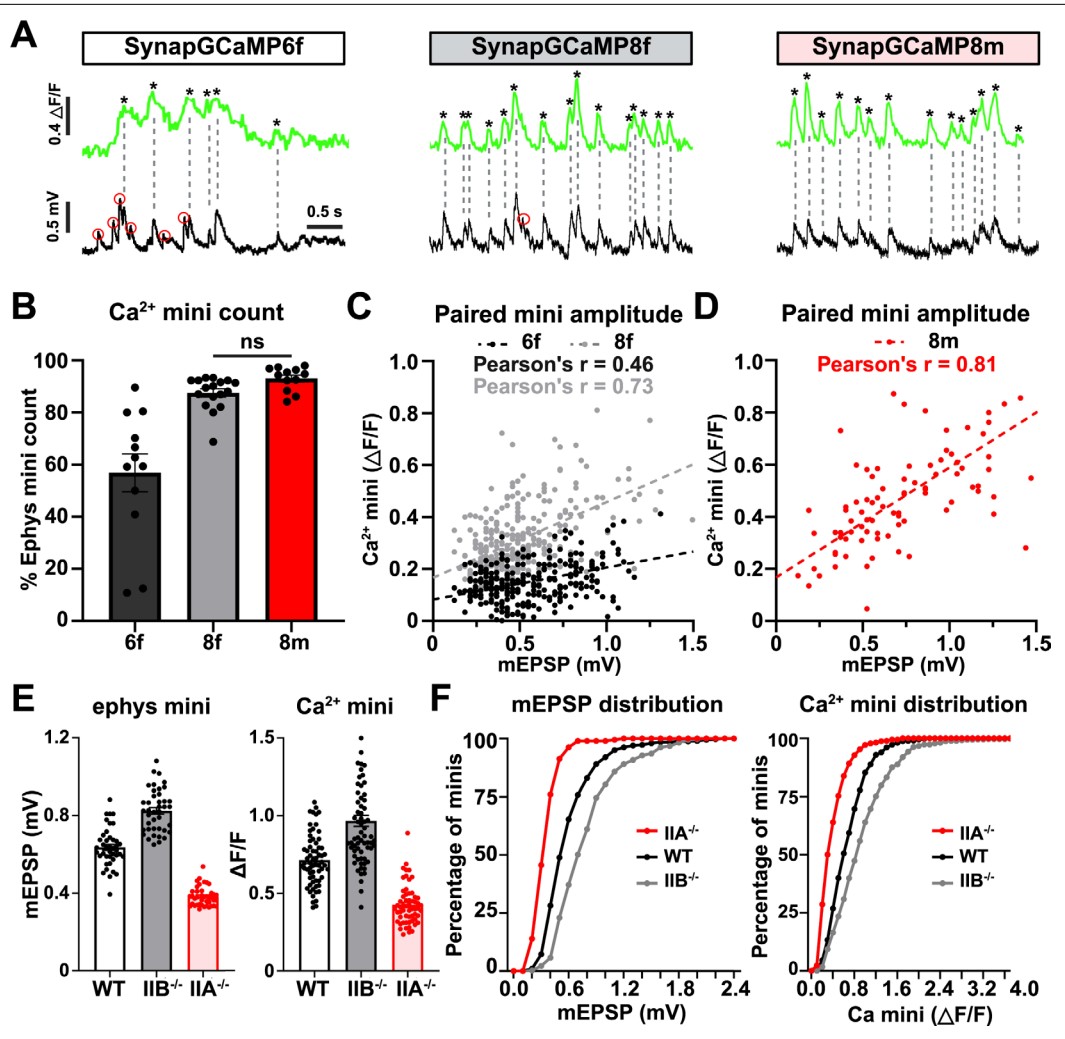

**Figure 8.** SynapGCaMP8m quantal resolution approaches that of electrophysiology. (**A**) Simultaneous recordings of quantal events at MN-Ib boutons using the indicated SynapGCaMP variant (green) and electrophysiology (black) after silencing MN-Is (*w;Is-GAL4/+;UAS-BoNT-C/SynapGCaMP*). Red circles indicate mEPSP events not captured by SynapGCaMP, and asterisks mark Ca²⁺ minis detected optically. SynapGCaMP8m captures quantal events with high sensitivity, comparable to electrophysiology. (**B**) Quantification of the proportion of quantal events captured by the indicated SynapGCaMP variant as a proportion of the total mEPSP events recorded by electrophysiology. SynapGCaMP6f detects only about half of electrophysiological events, while both SynapGCaMP8f and SynapGCaMP8m capture nearly all mEPSPs. (**C**) Scatter plot of paired miniature event amplitudes recorded simultaneously by SynapGCaMP6f and –8 f and electrophysiology. Each point represents a single matched event; Pearson's correlation coefficients (*r*=0.46 for 6 f and *r*=0.73 for 8 f) are indicated. (**D**) Scatter plot of paired miniature event amplitudes recorded simultaneously by SynapGCaMP8m and electrophysiology. Each point represents a single matched event; a strong linear relationship is observed (Pearson's *r*=0.81), indicating that optical signals scale proportionally with quantal amplitude. (**E**) Bar plots showing average mEPSP amplitudes (left) and ΔF/F amplitudes of quantal Ca²⁺ events (right) in the indicated genotypes (same conditions as in (**A**) but, including *GluRIIA* or *GluRIIB* mutant alleles). SynapGCaMP8m accurately resolves quantal size differences with similar resolution as the electrophysiological data, with quantal amplitudes in both datasets exhibiting the expected relationship (GluRIIA⁻/⁻<WT<GluRIIB⁻/⁻). Bars show mean ± SEM, dots represent individual boutons. (**F**) Cumulative frequency distributions of mEPSP amplitudes (left) and Ca²⁺ mini event amplitudes (right); each are significantly different using the Kolmogorov–Smirnov Test. See ***Supplementary file 1*** for full statistical details including *p*-values. Created with BioRender.com.

The online version of this article includes the following video for figure 8:

**Figure 8—video 1.** Example SynapGCaMP8m time-lapse imaging.
https://elifesciences.org/articles/107939/figures#fig8video1

and mEPSP recordings showed a high degree of temporal correspondence and sensitivity between mEPSPs and Ca²⁺ transients, with SynapGCaMP8m capturing the vast majority of electrophysiologically detected mEPSP events (*Figure 8A and B*). Indeed, mEPSP events that failed to register a detectable Ca²⁺ transient were most frequent with SynapGCaMP6f (~40%), rare with SynapGCaMP8f (~10%), and nearly absent with SynapGCaMP8m (~7%) (*Figure 8B*). Importantly, when quantal amplitudes from paired mEPSPs and corresponding Ca²⁺ transients were plotted, SynapGCaMP8m exhibited a strong linear correlation with electrophysiological recordings (Pearson's $r=0.810$, $R^2=0.656$; $p<0.0001$, n=216), demonstrating that the indicator reliably tracks quantal variability (*Figure 8D*). SynapGCaMP6f and SynapGCaMP8f also showed significant positive correlations, although weaker than SynapGCaMP8m (6 f: $r=0.458$, $R^2=0.210$, n=283; 8 f: $r=0.732$, $R^2=0.537$, n=298; all $p<0.0001$) (*Figure 8C*). Thus, SynapGCaMP8m provides a sensitive indicator of quantal events, approaching the sensitivity of electrophysiological recordings.

In our final set of experiments, we focused on benchmarking the ability of SynapGCaMP8m to resolve physiologically relevant differences in quantal events compared to electrophysiology. Specifically, we compared electrophysiological recordings and SynapGCaMP8m imaging of quantal events in three genotypes with known differences in quantal amplitudes: wild-type as the baseline control, *GluRIIA* mutants with diminished quantal amplitudes, and *GluRIIB* mutants with enlarged quantal events (*Han et al., 2023*). Electrophysiological recordings from each of the three genotypes confirmed the expected differences, with baseline mEPSPs from MN-Ib averaging ~0.64 mV, *GluRIIB* mutants ~0.82 mV, and *GluRIIA* mutants ~0.38 mV (*Figure 8E*), as shown in previous studies (*Han et al., 2023*; *Han et al., 2022*; *He et al., 2023*). Plotting the mEPSP distribution as a cumulative probability histogram also showed the expected distribution of these genotypes (*Figure 8F*). We then performed a similar analysis of quantal events imaged with SynapGCaMP8m. These results demonstrated near-identical differences in average quantal amplitude and distribution in quantal events imaged from wild-type, *GluRIIA*, and *GluRIIB* mutants (*Figure 8E and F*). In simultaneous SynapGCaMP8m imaging and current-clamp recordings, the vast majority of optical miniature events were accompanied by a corresponding mEPSP, with less than 5% of SynapGCaMP8m transients lacking a clearly detectable mEPSP; these may reflect very small quantal events that fall below the electrophysiological detection threshold (~0.2 mV). These data demonstrate that SynapGCaMP8m can capture physiologically relevant differences in quantal events with similar sensitivity as electrophysiology.

## Discussion

This study presents a significant advancement in our capacity to optically interrogate Ca²⁺ dynamics at synaptic compartments of the *Drosophila* neuromuscular junction, approaches that in principle can be extended to other systems. By systematically engineering and rigorously benchmarking a new suite of ratiometric GECIs – presynaptic Scar8m, active-zone targeted Bar8m, and postsynaptic SynapGCaMP8m – and complemented with the development of the CaFire automated analysis platform, we have overcome limitations of previous tools. These indicators provide a powerful toolkit for dissecting the Ca²⁺ signals that mediate and control synaptic transmission and plasticity, moving us closer to all-optical interrogation with similar resolution as electrophysiology. These results demonstrate that careful sensor design, incorporating the latest GCaMP8m variant and optimized ratiometric partners like mScarlet3, can yield probes with unprecedented performance in this model system.

Presynaptic Ca²⁺ imaging with Scar8m not only surpasses previous GECIs but also rivals, and in some respects exceeds, the capabilities of traditional synthetic Ca²⁺ indicators. While chemical dyes offer rapid kinetics (*Hendel et al., 2008*; *Lock et al., 2015*; *Macleod, 2012a*; *Lnenicka et al., 2006*; *Justs et al., 2022*; *Lu et al., 2016a*), their utility is often hampered by inconsistent loading/concentrations, lack of cell-type specificity, and challenges in ratiometric quantification within the confined space of presynaptic boutons. Indeed, traditional approaches to load dyes into fly motor neurons rely on cutting the nerve while bathed in solution, and waiting for the dye to diffuse the considerable distance to presynaptic terminals, where variation in dye concentration can confound analyses (*McMahon and Jackson, 2018*; *Macleod, 2012b*). Scar8m, targeted to synaptic vesicle pools at boutons via fusion to Synaptotagmin along with the ratiometric mScarlet3 motif, circumvents these issues, providing robust, genetically targeted, and stoichiometric and quantifiable Ca²⁺ measurements, spatially averaged and confined to individual boutons that outperform cytosolic GCaMPs (*Figure 3*). Furthermore, the superior sensitivity and relatively fast kinetics of Scar8f/m enabled the resolution of physiologically

meaningful differences in presynaptic $Ca^{2+}$ levels, such as those between MN-Ib and MN-Is terminals and the elevations associated with presynaptic homeostatic plasticity (*Li et al., 2021*). When directly comparing OGB-1 to Scar8f, we found superior sensitivity and kinetics (*Figure 3*), which, by extrapolation, should also compare favorably to other synthetic dyes like rhod and Fura (*Lu et al., 2016b*, *He et al., 2023*). Notably, Scar8f achieves a ΔF/F of approximately 0.32 vs 0.23 for OGB-1, while also showing a faster decay time constant (~40 ms for Scar8f vs. ~60 ms for OGB-1). Furthermore, the rise time performance of Scar8f/m is on par with that of OGB-1, with a single-AP rise time of ~12 ms compared to ~8 ms (*Figure 3*), with the caveat that scan speed can limit the accuracy of rise times. This capacity to faithfully report presynaptic $Ca^{2+}$ changes, previously accessible primarily via chemical dyes, underscores Scar8m's power as a tool for investigating presynaptic function with enhanced precision and reliability.

Equally significant is the performance of the postsynaptically targeted SynapGCaMP8m, which we have shown achieves a sensitivity for detecting quantal $Ca^{2+}$ events that approaches the gold standard of electrophysiological recordings. Previous optical methods often struggled to reliably capture these small, stochastic events, particularly in the case of glutamate receptor loss or perturbation (*Newman et al., 2017*; *Peled et al., 2014*; *Han et al., 2022*), and have never previously been benchmarked simultaneously with electrophysiological recordings to determine their fidelity. We found that SynapGCaMP6f failed to capture over 40% of mEPSP events, while the GCaMP8m version reliably detected over 90% of these events, even in cases of diminished amplitude due to *GluRIIA* loss. Leveraging the high responsivity of GCaMP8m enabled substantially improved peak amplitudes and kinetics. Furthermore, the strong correlation observed between the amplitudes of optical quantal events and their corresponding mEPSPs, along with the ability to accurately resolve known differences in quantal size in *GluRIIA* and *GluRIIB* mutants, serves to validate SynapGCaMP8m as a high-fidelity reporter of postsynaptic activity. Finally, we note the additional benefits of quantal imaging when compared to electrophysiology, where spatial discrimination of quantal events allows, for example, the ability to quantify input-specific events (see *Figure 8—video 1*). This opens exciting avenues for all-optical interrogation of quantal parameters, synaptic strength, and changes due to plasticity with subcellular spatial resolution.

Our investigation into capturing local $Ca^{2+}$ signals using an active zone-targeted, ratiometric GCaMP8m-based indicator, Bar8m, while ambitious, also yielded important insights despite the inherent technical challenges. Bar8m was able to sensitively report $Ca^{2+}$ changes at individual AZs following single AP electrical stimulation, as evidenced by the significant differences observed across AZs, expected by variation in $Ca_v2$ channel abundance shown in previous studies (*Cunningham et al., 2022*; *Gratz et al., 2019*; *Medeiros et al., 2024*). However, the Bar8f/m intensity signals showed relative consistency across AZs following single AP stimulation and imaged using resonant area scans. This finding, coupled with theoretical considerations of $Ca^{2+}$ diffusion rates (*Sinha et al., 1997*; *Helmchen et al., 1997*), suggests that within the temporal resolution of current GCaMP sensors, and likely synthetic dyes, $Ca^{2+}$ concentrations are likely to equilibrate across AZs within an individual bouton before distinct local concentration differences can be reliably resolved. There are a number of other potential confounds precluding a definitive conclusion about the ability of Bar8f/m to resolve local $Ca^{2+}$ changes, including challenges in resolving individual AZs and imaging speeds. Indeed, more promising results were observed using line scanning, with kHz speeds, of Bar8m, which appeared to capture a higher degree of AZ-specific differences (*Figure 6E*). These observations are critical for interpreting data acquired from AZ-targeted GECIs and guiding future efforts to capture localized $Ca^{2+}$ nanodomain dynamics, likely requiring sensors with even faster kinetics and/or alternative imaging modalities.

The suite of next-generation GECIs, particularly Scar8m and SynapGCaMP8m, coupled with the CaFire analysis pipeline, represents a substantial toolkit for studying synaptic $Ca^{2+}$ signaling. At the *Drosophila* NMJ, these tools provide researchers with unprecedented optical access to both presynaptic $Ca^{2+}$ influx driving neurotransmitter release and postsynaptic $Ca^{2+}$ transients reflecting quantal events, with performance characteristics that rival traditional, more invasive methods. The ability to perform ratiometric, genetically targeted imaging with such high sensitivity and temporal resolution will help to accelerate discoveries into the molecular mechanisms of synaptic transmission, plasticity, and disease etiology. More generally, the strategies used here in the fly system can inspire similar approaches to be employed in other systems. Future studies leveraging these advanced sensors can

now tackle complex questions regarding the spatial and temporal dynamics of Ca²⁺ signaling within synaptic compartments, the modulation of these signals during various forms of plasticity, and their dysregulation in models of neurological disorders, further cementing the *Drosophila* NMJ as a premier system for fundamental synaptic research.

## Materials and methods

**Key resources table**

| Reagent type (species) or resource | Designation | Source or reference | Identifiers | Additional information |
|---|---|---|---|---|
| Antibody | mouse monoclonal anti-DLG | Developmental Studies Hybridoma Bank (DSHB) | RRID:AB_528203 | (1:50) |
| Antibody | mouse monoclonal anti-BRP | DSHB | RRID:AB_2314866 | (1:100) |
| Antibody | chicken polyclonal anti-GFP | Aves Lab | RRID:AB_2307313 | (1:1000) |
| Antibody | rabbit polyclonal anti-GluRIIC | PMID:29748610 | | (1:2000) |
| Antibody | rabbit polyclonal anti-Syt | PMID:12110842 | | (1:2000) |
| Antibody | Alexa Fluor 488 conjugated Donkey anti-chicken secondary antibody | Jackson ImmunoResearch | RRID:AB_2340375 | (1:400) |
| Antibody | Alexa Fluor 488 conjugated Goat anti-Horseradish Peroxidase | Jackson ImmunoResearch | RRID:AB_2338965 | (1:400) |
| Antibody | Alexa Fluor 594 conjugated Goat anti-mouse secondary antibody | Jackson ImmunoResearch | RRID:AB_2340854 | (1:400) |
| Antibody | Alexa Fluor 647 conjugated Donkey anti-mouse secondary antibody | Jackson ImmunoResearch | RRID:AB_2340862 | (1:400) |
| Antibody | Alexa Fluor 647 conjugated Donkey anti-rabbit secondary antibody | Jackson ImmunoResearch | RRID:AB_2492288 | (1:400) |
| Antibody | Cy3-conjugated Donkey anti-rabbit secondary antibody | Jackson ImmunoResearch | RRID:AB_2307443 | (1:400) |
| Antibody | DyLight 405 conjugated Donkey anti-mouse secondary antibody | Jackson ImmunoResearch | RRID:AB_2340839 | (1:400) |
| Antibody | STAR RED conjugated secondary antibodies | Abberior | Cat#:STRED-1002 | (1:200) |
| Chemical compound, drug | Oregon Green 488 BAPTA-1 dextran, Potassium Salt, 10,000 MW, Anionic | Thermo Fisher Scientific (Invitrogen, Molecular Probes) | Cat#:O6798 | (5 mM) |
| Genetic reagent (*Drosophila melanogaster*) | UAS-Syt::mScarlet::GCaMP8f (Scar8f) | PMID:34851664 | | |
| Genetic reagent (*D. melanogaster*) | UAS-Syt::mScarlet3::GCaMP8m (Scar8m) | This paper | | See Materials and methods 'Molecular biology' section |
| Genetic reagent (*D. melanogaster*) | UAS-Syt::GCaMP6s | BDSC | RRID:BDSC_64414 | |
| Genetic reagent (*D. melanogaster*) | RSET-GCaMP8m | BDSC | RRID:BDSC_605073 | |
| Genetic reagent (*D. melanogaster*) | SynapGCaMP6f | PMID:28285823 | | |
| Genetic reagent (*D. melanogaster*) | SynapGCaMP8f | PMID:35993544 | | |
| Genetic reagent (*D. melanogaster*) | SynapGCaMP8m | This paper | | |
| Genetic reagent (*D. melanogaster*) | UAS-BRP::mCherry::GCaMP6s | PMID:28658618 | | |

*Continued on next page*

*Continued*

| Reagent type (species) or resource | Designation | Source or reference | Identifiers | Additional information |
|---|---|---|---|---|
| Genetic reagent (*D. melanogaster*) | UAS-BRP::mScarlet::GCaMP8f (Bar8f) | This paper | | See Materials and methods 'Molecular biology' section |
| Genetic reagent (*D. melanogaster*) | UAS-BRP::mScarlet3::GCaMP8m (Bar8m) | This paper | | See Materials and methods 'Molecular biology' section |
| Genetic reagent (*D. melanogaster*) | OK6-GAL4 | PMID:11856529 | | |
| Genetic reagent (*D. melanogaster*) | OK319-GAL4 | PMID:7857643 | | |
| Genetic reagent (*D. melanogaster*) | UAS-BoNT-C | PMID:35993544 | | |
| Genetic reagent (*D. melanogaster*) | R27E09-GAL4 (Is-Gal4) | BDSC | RRID:BDSC_49227 | |
| Strain, strain background (*D. melanogaster*) | w[1118] | BDSC | RRID:BDSC_5905 | |
| Genetic reagent (*D. melanogaster*) | GluRIIA[pv3] | PMID:37436892 | | |
| Genetic reagent (*D. melanogaster*) | GluRIIB[sp5] | PMID:37436892 | | |
| Recombinant DNA reagent | pJFRC81-Syt::mScarlet::GCaMP8f (Scar8f) | PMID:34851664 | | |
| Recombinant DNA reagent | pJFRC81-Syt::mScarlet3::GCaMP8m (Scar8m) | This paper; GenBank | PZ014175 | See Materials and Methods 'Molecular biology' section |
| Recombinant DNA reagent | pACU2-BRPs_mScarlet_GCaMP8f (Bar8f) | This paper; GenBank | PZ014172 | See Materials and methods 'Molecular biology' section |
| Recombinant DNA reagent | pACU2-BRPs_mScarlet_GCaMP8m (Bar8m) | This paper; GenBank | PZ014173 | See Materials and methods 'Molecular biology' section |
| Recombinant DNA reagent | pC2A-MHC-CD8-GCaMP8f-Sh (SynapGCaMP8f) | PMID:35993544 | | |
| Recombinant DNA reagent | pC2A-MHC-CD8-GCaMP8m-Sh (SynapGCaMP8m) | This paper; GenBank | PZ014174 | See Materials and methods 'Molecular biology' section |
| Software, algorithm | ImageJ | https://imagej.net/ | version:1.8.0 | |
| Software, algorithm | NIS-Elements software | Nikon Instruments | version:5.41.02 (Build 1711) | |
| Software, algorithm | Huygens Essential | Scientific Volume Imaging | version:25.04 | |
| Software, algorithm | Mini Analysis | Synaptosoft | version:6.0.7 | |
| Software, algorithm | Axon pCLAMP Clampfit | Molecular Devices | version:10.7 | |
| Software, algorithm | Clampex | Molecular Devices | version:10.7 | |
| Software, algorithm | GraphPad Prism | GraphPad | version:10.0.1 | |
| Software, algorithm | Jupyter Notebook | Anaconda | version:6.0.1 | |
| Software, algorithm | Python | https://www.python.org/ | version:3.10.11 | |
| Software, algorithm | Excel | Microsoft | version:2021 | |
| Software, algorithm | CaFire | This paper | version:2.2.1 | https://github.com/linj7/CaFire; *Lin, 2026* |

## Fly stocks

*Drosophila* stocks were raised at 25 °C using standard molasses food. Unless otherwise specified, the *w¹¹¹⁸* strain was used as the wild-type control, as this is the genetic background in which all genotypes were bred. All experiments were performed on *Drosophila* third-instar larvae of both sexes unless otherwise noted. See the Key resources table for a full list of all fly stocks and their sources used in this study.

## Molecular biology

To generate the transgenic constructs used in this study, we built upon previously established tools. First, SynapGCaMP8m was engineered by modifying the previous SynapGCaMP8f construct (*Han et al., 2022*), which was in turn based on the original SynapGCaMP6f construct (*Newman et al., 2017*). Gibson assembly reactions were used to convert the few relevant amino acid differences between GCaMP8f and GCaMP8m (*Zhang et al., 2023*). Similarly, UAS-Syt::mScarlet3::GCaMP8m (Scar8m) was generated using similar approaches to modify the previously described UAS-Syt::mScarlet::GCaMP8f plasmid (*Li et al., 2021*), changing both the GCaMP8f and mScarlet sequences to GCaMP8m and mScarlet3 (*Gadella et al., 2023*). Finally, UAS-BrpS::mScarlet::GCaMP8f (Bar8f) and UAS-BrpS::mScarlet3::GCaMP8m (Bar8m) were engineered based on the previous UAS-BrpS::mCherry::GCaMP6s transgene (*Kiragasi et al., 2017*), where the GCaMP6s sequence was replaced with GCaMP8f or GCaMP8m, and the mCherry tag was substituted with mScarlet or mScarlet3. SynapGCaMP transgenes were inserted randomly through p-element transposition, while all other transgenes were inserted into the attP40 (II) or VK27 (III) sites by BestGene Inc (Chino Hills, CA). All constructs were verified by Sanger sequencing prior to injection, and transgenic fly stocks were established and maintained under standard laboratory conditions.

## Immunocytochemistry

Third-instar larvae were dissected in ice-cold 0 $Ca^{2+}$ HL-3 and immunostained as described (*Li et al., 2021*; *Han et al., 2023*). Briefly, larvae were either fixed in 100% ice-cold methanol for 5 min or PFA for 10 min followed by washing with PBS containing 0.1% Triton X-100 (PBST). Samples were blocked with 5% Normal Donkey Serum for 1 hr and incubated with primary antibodies overnight at 4 °C. Preparations were washed for 10 min thrice in PBST, incubated with secondary antibodies for 2 hr at room temperature, washed thrice again in PBST, and equilibrated in 70% glycerol. Prior to imaging, samples were transferred in VectaShield (Vector Laboratories, Burlingame, CA) and mounted on glass cover slides. Details of all antibodies and sources are listed in the Key resources table.

## Electrophysiology

All dissections and electrophysiological recordings were performed as described (*Kikuma et al., 2019*) in modified hemolymph-like saline (HL-3) containing (in mM): 70 NaCl, 5 KCl, 10 $MgCl_2$, 10 $NaHCO_3$, 115 Sucrose, 5 Trehalose, 5 HEPES, pH = 7.2, and $CaCl_2$ at 0.4 mM unless otherwise specified. Internal guts, brain, and the ventral nerve cord were subsequently removed to achieve fully dissected preparations. Recordings were carried out on an Olympus BX61 WI microscope stage equipped with a 40 x/0.8 NA water-dipping objective and acquired using an Axoclamp 900 A amplifier (Molecular Devices). All recordings were conducted on abdominal muscle 6, segment A3 of third-instar larvae of both sexes. Data were acquired from cells with an initial resting potential between –60 and –80 mV and input resistances >5 MΩ. Miniature excitatory postsynaptic potentials (mEPSPs) were recorded without any stimulation and low-pass filtered at 1 kHz. The mEPSPs for each sample were recorded for 60 s, analyzed with MiniAnalysis (Synaptosoft), and the average mEPSP amplitude for each NMJ was calculated. Excitatory postsynaptic potentials (EPSPs) were recorded by delivering 20 electrical stimuli at 0.5 Hz with 0.5 ms duration to motor neurons using an ISO-Flex stimulus isolator (A.M.P.I.) with stimulus intensities set to avoid eliciting multiple EPSPs.

## Confocal imaging

Dissections and live $Ca^{2+}$ imaging was performed as described (*Chien et al., 2025*; *Chen et al., 2024*) on muscle 6 NMJs. Wandering third-instar larvae were dissected and imaged in 1.8 mM $Ca^{2+}$ HL3 saline. $Ca^{2+}$ imaging was conducted using a Nikon A1R resonant scanning confocal microscope equipped with a 60 x/1.0 NA water-immersion objective (refractive index 1.33). GCaMP signals were acquired

using the FITC/GFP channel (488 nm laser excitation; emission collected with a 525/50 nm band-pass filter), and mScarlet/mCherry signals were acquired using the TRITC/mCherry channel (561 nm laser excitation; emission collected with a 595/50 nm band-pass filter). ROIs focused on terminal boutons of MN-Ib or -Is motor neurons. For both channels, the confocal pinhole was set to a fixed diameter of 117.5 μm (approximately three Airy units under these conditions), which increases signal collection while maintaining adequate optical sectioning. Images were acquired as 256×64 pixel frames (two 12-bit channels) using bidirectional resonant scanning at a frame rate of ~118 frames/s; the scan zoom in NIS-Elements was adjusted so that this field of view encompassed the entire neuromuscular junction and was kept constant across experiments. In ratiometric recordings, the 488 nm (GCaMP) and 561 nm (mScarlet) channels were acquired in a sequential dual-channel mode using the same bidirectional resonant scan settings: for each time point, a frame was first collected in the green channel and then immediately in the red channel, introducing a small, fixed frame-to-frame temporal offset while preserving matched spatial sampling of the two channels. Resonant area scans of Bar8f and Bar8m responses were acquired at ~60 or~120 frames/s with a scanning area of 256×32–64 pixels, using a zoom factor of 8x. Because Bar8f was dimmer, the confocal image quality setting was increased from level 1–2, which reduced the scanning frame rate. To resolve rapid $Ca^{2+}$ transients at individual AZs, we additionally performed single-channel Galvano line-scan imaging of Bar8m. A brief resonant area image was first acquired to localize BRP-positive puncta, and a one-dimensional scan line was then positioned through the center of each selected AZ. Line-scan images were acquired at ~1000 frames/s sampling rate. Presynaptic and active zone $Ca^{2+}$ imaging was conducted for 15 s per acquisition while delivering electrical stimulation at 1 Hz (1 ms pulse duration) or at 5 Hz and 10 Hz (five pulses per 2 s burst). Action potentials were evoked by extracellular stimulation of the segmental motor nerve using a glass suction electrode. Suction electrodes were pulled from TW120-4 glass capillaries on a Sutter P-97 puller and fire-polished to a final tip opening of ~5–10 μm, then filled with HL3 saline. The cut motor nerve innervating the imaged muscle was gently drawn into the electrode, and electric stimulation was delivered through a pulse stimulator coupled to an isolated stimulator (Iso-Flex, controlled by a Master-9 pulse generator) (*Chen et al., 2024*). To prevent muscle contraction during imaging, 7 mM L-Glutamic acid monosodium salt (Sigma-Aldrich, Cat# G5889) was added to the HL3 saline. In contrast, postsynaptic $Ca^{2+}$ imaging was performed for 60 s without electrical stimulation, as only quantal events were monitored. Data was collected from at least three biological replicates per genotype.

Time-lapse videos were stabilized and bleach-corrected prior to analysis, which visibly reduced frame-to-frame motion and intensity drift. In the presynaptic and active-zone mScarlet channel, a bleaching factor of ~1.15 was observed during the 15 s recording. This bleaching can be corrected using the 'Bleaching correction' tool in Huygens SVI. For presynaptic and active-zone GCaMP signals, there was minimal bleaching over these short imaging periods. Therefore, the bleaching correction step for GCaMP was skipped. Both GCaMP and mScarlet channels were processed using the default settings in the Huygens SVI 'Deconvolution Wizard' (with the exception of the bleaching correction option). Deconvolution was performed using the CMLE algorithm with the Huygens default stopping criterion and a maximum of 30 iterations, such that the algorithm either converged earlier or, if convergence was not reached, was terminated at this 30-iteration limit; no other iteration settings were used across the GCaMP series. ROIs were drawn on the processed images using Fiji ImageJ software, and mean fluorescence time courses were extracted for the GCaMP and mScarlet channels, yielding $F_{GCaMP}(t)$ and $F_{mScarlet}(t)$. F(t)s were imported into CaFire with GCaMP assigned to Channel #1 (signal; required) and mScarlet to Channel #2 (baseline/reference; optional). If desired, the mScarlet signal could be smoothed in CaFire using a user-specified moving-average window to reduce high-frequency noise. In CaFire's ΔR/R mode, the per-frame ratio was computed as $R(t)=F_{GCaMP}(t)/F_{mScarlet}(t)$; a baseline ratio R0 was estimated from the pre-stimulus period, and the final response was reported as $\Delta R/R(t)=[R(t)-R0]/R0$, which normalizes GCaMP signals to the co-expressed mScarlet reference and thereby reduces variability arising from differences in sensor expression level or illumination across AZs (see CaFire section in methods for details).

Confocal images of fixed tissue were acquired with a 100 x APO 1.45 NA oil immersion objective using separate channels with four laser lines (405 nm, 488 nm, 561 nm, and 647 nm) as described (*Perry et al., 2017*). Z-stacks were obtained on the same day using identical gain and laser power settings with z-axis spacing between 0.13 and 0.2 μm and pixel size of 0.06 μm for all samples within

an individual experiment, from at least eight NMJs acquired from at least four different animals. Raw confocal images were deconvolved with SVI Huygens Essential 22.10 using built-in Express settings. All confocal images shown in the figures, including *Figure 1*, were deconvolved in Huygens using this CMLE-based workflow prior to maximum-intensity projection and display. Maximum intensity projections were created for quantitative image analysis using the general analysis toolkit of NIS Elements software.

## Benchmarking OGB-1 vs. Scar8f

For OGB-1 imaging, wild-type motor neuron terminals were forward filled with Oregon Green 488 BAPTA-1 (OGB-1; 10,000 MW anionic dextran, batch-specific Kd = 1,180 nM, Molecular Probes) as described (*Macleod et al., 2002*). Larvae were rinsed in distilled water, transferred to Schneider's *Drosophila* medium, and pinned dorsal-side up on Sylgard-coated dishes with Minutien pins. A midline incision was made along the dorsal surface, the body wall was pinned open to generate a fillet preparation, and internal organs were removed to expose the ventral nerve cord and segmental nerves. For forward filling, segmental nerves were severed near the ventral ganglion and the cut ends from hemisegment 4 were drawn into a glass suction pipette; a bolus of 5 mM OGB-1 in distilled water was ejected onto the nerve ends in an equal volume of Schneider's medium, left in place for 30–60 min, then replaced with Schneider's medium in the pipette to serve as the cathode for stimulation. OGB-1 was allowed to transport to NM6/7-Ib terminals for 4–5 hr at room temperature, with Schneider's medium exchanged hourly. Immediately prior to imaging, Schneider's medium was replaced with hemolymph-like 6 (HL6) saline containing 2 mM $CaCl_2$ and 7 mM L-glutamic acid for at least 15 min. $Ca^{2+}$ imaging was performed on an upright confocal laser-scanning microscope (Olympus Fluoview 4000) equipped with a LUMPlanFl 60x/0.90 W objective. GCaMP8f or OGB-1 were excited at 488 nm using 10% and 5% laser power, respectively, and emission was collected through a maximized pinhole using a grating and slit set to 510–550 nm. mScarlet was excited at 561 nm (2% power) and emission collected at 570–620 nm. Images were acquired with a 2x digital zoom (190 nm/pixel) over a 512×48 pixel frame at 3.295 ms per frame. For each recording, 1 s of baseline was collected prior to nerve stimulation, followed by 14 s of imaging to capture 10 isolated single AP responses and a subsequent train of 17 APs. Nerves innervating segment 4 were stimulated using the silver wire in the suction pipette as the cathode and a bath wire as the anode, driven by an isolated pulse stimulator (DS2A, Digitimer) controlled by a Master-8 (A.M.P.I.). Stimuli were 1.5 V, 0.3 ms pulses delivered as 10 APs at 1 Hz, followed after a 2 s delay by 17 APs at 21.3 Hz. Only preparations with stable baseline fluorescence and reproducible responses were analyzed. For each indicator, seven larvae were imaged, with the full stimulus trial (10 @ 1 Hz followed by 17 @ 21.3 Hz) repeated three times per larva. Fluorescence was quantified from ROIs surrounding 1–2 non-terminating MN6/7-Ib boutons; background-subtracted stimulus fluorescence ($F_{stim}$) was obtained using cellSens FV software (v4.1.1, Evident Scientific) and further processed in Microsoft Excel. Bleaching (typically ~8% and 12% per 10 s for GCaMP8f and OGB-1, respectively) was corrected by fitting a linear trend to the average pre-stimulus fluorescence in each 100 ms window preceding the 10 APs at 1 Hz and adjusting each F_stim according to its time and the slope of this line. ΔF/F was then calculated as $(F_{stim}-F_{rest})/F_{rest}$, where $F_{rest}$ is the mean fluorescence during the 100 ms preceding the first AP. mScarlet fluorescence was not used when benchmarking Scar8f against OGB-1 because a red dextran fluorophore was not co-loaded in constant proportion with OGB-1. For single-AP analysis, three trials per preparation were averaged, and the 10 responses were aligned so that the first rise in $F_{stim}$ occurred 1.648 ms (half a sampling interval) after the stimulus. Amplitude was obtained from the intercept at time 0 of a single-exponential fit ($\Delta F/F = \Delta F/F = A_o e^{-bt}$) to data from 14.828 to 499.193 ms (147 points), and the decay time constant $\tau$ was calculated as $1000 \text{ ms}/-b$. For train responses (17 @ 21.3 Hz), trials were averaged after alignment to the first rise in $F_{stim}$; amplitude was taken as the peak ΔF/F after the final AP, the facilitation index was computed as the ratio of the final to the first response, and $\tau$ was derived from a single-exponential fit to 1 s (304 points) of data following the peak.

## Widefield $Ca^{2+}$ imaging

Widefield $Ca^{2+}$ imaging was conducted on muscle 4 NMJs using a Zeiss Axio Examiner A1 upright fixed-stage microscope equipped with a pco.panda 4.2 sCMOS camera (Excelitas) in 1.8 mM $Ca^{2+}$ HL3 saline. High-power 470 nm LED (Thorlabs, M470L4) was used for illumination with a 63x/1.0 NA

water-dipping lens, with image acquisition controlled using Nikon NIS-Elements software. Regions of interest (ROIs) were defined to encompass terminal boutons of MN-Ib branches on muscle 4. Images were acquired at 256×256 pixels with a frame rate of 100 fps. GCaMP was imaged using a GFP filter cube (LED excitation ~470/40 nm, emission ~525/50 nm). Time-series imaging was performed for 60 s without external stimulation to capture spontaneous quantal Ca²⁺ events. Under these wide-field imaging conditions, SynapGCaMP signals exhibited only modest photobleaching, typically a ~2–5% decrease in baseline fluorescence over a 60 s acquisition. The modest bleaching was either corrected using the 'bleaching correction' function in Huygens SVI or accounted for during ΔF/F quantification in the CaFire analysis pipeline using a sliding-baseline procedure (see CaFire methods). For simultaneous electrophysiological recordings, Clampex software (Molecular Devices) was employed to perform current-clamp recordings in gap-free mode. A digital TTL trigger signal from Clampex synchronized the imaging acquisition in NIS-Elements, ensuring precise temporal alignment between electrophysiological and imaging data. Data was collected from a minimum of six biological replicates per genotype.

## STED imaging

STED super-resolution microscopy was performed as previously described (*He et al., 2023*; *Chien et al., 2025*). Briefly, STED imaging was performed with an Abberior STEDYCON system mounted on a Nikon Eclipse FN1 upright microscope. The system is equipped with four excitation lasers (640, 561, 488, and 405 nm), a pulsed STED laser at 775 nm, and three avalanche photodiode detectors operating in a single photon counting mode. Alexa Fluor 488-conjugated anti-HRP was used to locate NMJs, whereas Abberior STAR Red and Alexa Fluor 594 secondary dyes were used for the STED channels. Depletion was performed at 775 nm with time-gated detection set to open from 1 ns to 7 ns after each excitation pulse. Emission was detected with two avalanche photodiodes (band-pass filters: 600±25 nm for STAR RED and 675±25 nm for Alexa Fluor 594). Images were acquired sequentially with a pixel dwell time of 10 μs and 3x line accumulation. Multichannel z-stack STED images were acquired using a 100x Nikon Plan APO 1.45 NA oil immersion objective with a 20 nm fixed pixel size at 130 nm steps, which yields an effective lateral resolution of ~60 nm. Each image covered one to two boutons (25–64 μm²) on muscle 4 of segment A3. Raw STED images were corrected for thermal drift and channel crosstalk, then deconvolved in SVI Huygens software Essential 24.04.0 (Scientific Volume Imaging B.V.) using theoretical STED point-spread and default iteration settings of the Good's MLE algorithm.

## CaFire program

CaFire is a software tool designed for the analysis and processing of Ca²⁺ imaging data, compatible with both raw intensity data and pre-normalized signals (e.g. ΔF/F or ΔR/R). The graphical user interface (GUI) of CaFire is implemented with the Python Tkinter library, while the visualization panel is powered by Matplotlib. The method of automated peak detection utilizes signal processing functions from SciPy (*Virtanen et al., 2020*). Peaks that are not detected automatically can be selected manually. CaFire identifies the nearest peak to the location where the user clicks. For peak quantification analysis, CaFire calculates rise and decay properties by performing curve fitting on data preceding and following the peak, respectively. The curve fitting is performed using the Levenberg–Marquardt gradient descent algorithm, implemented in the curve fit function of SciPy (*Virtanen et al., 2020*). The rise time constant is calculated by applying an exponential fit to the rise phase (*Shemesh et al., 2020*). The decay is modeled using a natural exponential decay function: $y = y_{peak} \cdot e^{-t/\tau_{decay}}$ (*Fleming et al., 2021*).

After motion correction/processing, ROIs were drawn in Fiji and mean-intensity time series were exported as numeric tables. The traces were organized in an Excel (.xlsx) file, with the first row containing column headers (text labels) and all subsequent rows containing numeric data only. CaFire directly imports .xlsx tables and requires at least two columns per trace: time (or frame number) on the x-axis and intensity on the y-axis (additional ROIs can be placed in separate columns with distinct headers). If only frame numbers are provided, rise and decay tau results need to be converted to time separately by multiplying the fitted tau (in frames) by the frame interval (Δt=1000 ms/frame rate). CaFire supports both single- and dual-channel imaging: for all analyses, GCaMP traces are imported as Channel #1 (signal; required), and for dual-channel datasets the mScarlet reference is additionally

imported as Channel #2 (baseline/reference; optional). For single-channel recordings, CaFire can either accept user-supplied ΔF/F traces or compute ΔF/F from raw fluorescence: in the latter case, CaFire first estimates a time-varying baseline $F_0(t)$ using a sliding-window percentile method, with the window length, percentile, and time range user-configurable in the GUI (unless otherwise noted, we used a 101-point window, i.e., 50 frames before and after each time point, and the $30^{th}$ percentile of values within that window to define $F_0$). Fluorescence signals are then expressed as $\Delta F/F(t) = [F(t) - F_0(t)] / F_0(t)$, which we report throughout the manuscript as ΔF/F. For ratiometric datasets, CaFire allows users to analyze either precomputed ΔR/R traces or to generate ΔR/R from raw two-channel fluorescence data. ROIs are drawn in Fiji/ImageJ and mean fluorescence time courses are extracted for the green GCaMP channel and the red mScarlet channel, yielding $F_{GCaMP}(t)$ and $F_{mScarlet}(t)$, which are imported into CaFire as the signal and reference channels, respectively. If desired, the mScarlet signal can be smoothed in CaFire using a user-specified moving-average window (e.g. eight frames at ~118 fps) to reduce high-frequency noise without altering the underlying ratio dynamics. In ΔR/R mode, CaFire computes a per-frame ratio $R(t)=F_{GCaMP}(t)/F_{mScarlet}(t)$ and estimates a baseline ratio $R_0(t)$ using the same sliding-window percentile procedure; for Bar8m active-zone analysis, we used a 101-point window and the lowest 10% of ratio values within each window to define $R_0$. Ratiometric responses are then reported as $\Delta R/R(t) = [R(t) - R_0(t)]/R_0(t)$. CaFire can plot raw, ΔF/F, or ratiometric traces, and $F_0(t)$ or $R_0(t)$ can be visualized in the raw-data view for quality control. The source code and a Windows executable, together with additional instructions (including ImageJ workflow instructions, an Excel template, CaFire loading screenshots, and a brief step-by-step usage guide), are available for download on CaFire's GitHub repository (https://github.com/linj7/CaFire; *Lin, 2026*) and has been archived with a DOI at Zenodo: 10.5281/zenodo. 17563451. The software is released under the MIT License.

## Statistical analysis

Data were analyzed using NIS Elements software (Nikon), MiniAnalysis (Synaptosoft), SVI Huygens Essential (version 22.10), CaFire (https://github.com/linj7/CaFire; *Lin, 2026*), GraphPad Prism (version 10.0), and Microsoft Excel software (version 16.22). For comparisons between two groups, we used unpaired two-tailed Student's t-tests. For comparisons among more than two groups, we used one-way ANOVA followed by Tukey's multiple comparisons test. Cumulative distributions of mEPSP amplitudes and Ca²⁺ mini amplitudes were compared using two-sample Kolmogorov–Smirnov tests. Pearson correlation coefficients and linear regression were used to assess relationships between optical and electrophysiological miniature event amplitudes, as well as between AZ size and Ca²⁺ signal amplitude. When outlier analysis was required (e.g. analyzing AZs), outliers were identified using the interquartile range (IQR) method. To quantify AZ-to-AZ variability, we performed paired two-tailed t-tests between every pair of active zones within the same preparation, comparing peak responses across stimuli. For Bar8m line-scan recordings, ~79% of AZ pairs were significantly different at $\alpha<0.05$, whereas for Bar8f area-scan recordings, only ~8% of AZ pairs were significant, indicating substantially greater AZ-to-AZ diversity when measured at high temporal resolution. For visual marking of outlying AZs in the scatter plots, we computed the interquartile range (IQR; Q1–Q3) from the individual responses pooled across AZs (15 stimuli per AZ, 30 AZs per genotype) and highlighted AZs whose mean response lay above Q3 or below Q1. The IQR was used solely as a robust measure of dispersion for visualization, and not as a formal hypothesis test. The fraction of significant AZ pairs is reported on the corresponding plots. Throughout the manuscript, n refers to the number of animals (biological replicates). All Ca²⁺ imaging and electrophysiological recordings were performed at abdominal muscle 6, segment A3 of third-instar larvae. For imaging experiments, each larva contributed 1–2 NMJs in total; at each NMJ, we typically imaged 2–3 terminal boutons and acquired 2–3 time-series recordings from different boutons. For the standard stimulation protocol, 1 ms pulses were delivered at 1 Hz during a 15 s imaging epoch (yielding ~14–15 stimuli per trial). In all figures, error bars indicate ± SEM, with the following statistical significance: $p<0.05$ (*), $p<0.01$

(\*\*), $p<0.001$ (\*\*\*), $p<0.0001$ (\*\*\*\*); ns = not significant. Additional statistical details for all experiments are summarized in *Supplementary file 1*.

## Acknowledgements

We thank Greg Macleod (Tulane University, New Orleans, LA, USA), Martin Muller (University of Zurich, Zurich, Switzerland), and Igor Delvendahl (University of Freiburg, Freiburg, Germany) for useful discussions about $Ca^{2+}$ imaging and interpretations. We acknowledge the Developmental Studies Hybridoma Bank (Iowa, USA) for antibodies used in this study and the Bloomington *Drosophila* Stock Center for fly stocks (NIH P40OD018537). This work was supported by grants from the National Institutes of Health, NINDS awards to GM (NS123377) and DD (NS091546 and NS126654).

## Additional information

### Competing interests

Dion K Dickman: Reviewing editor, eLife. The other authors declare that no competing interests exist.

### Funding

| Funder | Grant reference number | Author |
|---|---|---|
| National Institute of Neurological Disorders and Stroke | NS123377 | Gregory T Macleod |
| National Institute of Neurological Disorders and Stroke | NS091546 | Dion K Dickman |
| National Institute of Neurological Disorders and Stroke | NS126654 | Dion K Dickman |

The funders had no role in study design, data collection and interpretation, or the decision to submit the work for publication.

### Author contributions

Jiawen Chen, Data curation, Formal analysis, Validation, Investigation; Junhao Lin, Data curation, Software; Kaikai He, Formal analysis, Methodology; Luyi Wang, Chengjie Qiu, Jasmine M Wheeler, Catherine M Daly, Data curation; Yifu Han, Data curation, Methodology; Gregory T Macleod, Formal analysis, Supervision, Funding acquisition, Writing – review and editing; Dion K Dickman, Conceptualization, Supervision, Funding acquisition, Methodology, Writing – original draft, Project administration, Writing – review and editing

### Author ORCIDs

Jiawen Chen ORCID https://orcid.org/0009-0003-6582-6559
Yifu Han ORCID https://orcid.org/0000-0002-1201-654X
Chengjie Qiu ORCID https://orcid.org/0000-0002-5382-8325
Dion K Dickman ORCID https://orcid.org/0000-0003-1884-284X

Reviewer #1 (Public review): https://doi.org/10.7554/eLife.107939.3.sa1
Reviewer #2 (Public review): https://doi.org/10.7554/eLife.107939.3.sa2
Reviewer #3 (Public review): https://doi.org/10.7554/eLife.107939.3.sa3
Author response https://doi.org/10.7554/eLife.107939.3.sa4

## Additional files

### Supplementary files

Supplementary file 1. Absolute values and statistical comparisons for $Ca^{2+}$ imaging and

electrophysiology data. This table reports the full statistical details and properties for data presented in the indicated figures, including *p*-values, mean ± SEM, sample sizes (n), and genotypes for all conditions tested. Ca²⁺ imaging parameters include ΔF/F (or ΔR/R), rise time ($\tau_{rise}$), and decay time ($\tau_{decay}$) constants. Electrophysiological parameters include mEPSP amplitude, EPSP amplitude, quantal content (QC), input resistance, and resting potential. *p*-values from one-way ANOVA with Tukey's multiple comparison test are shown for key contrasts between genotypes and indicators. Data for outlier analysis, correlation analysis and linear regression analyses are included where applicable.

MDAR checklist

## Data availability

All relevant data is included in the publication, primarily in *Supplementary file 1*. All fly stocks and molecular constructs generated in this study will be shared upon request. We are also in the process of depositing newly generated Scar8f/m, Bar8f/m, and SynapGCaMP sensors to the Bloomington Drosophila Stock Center for public dissemination. The source code for the CaFire analysis software is publicly available on GitHub at https://github.com/linj7/CaFire (*Lin, 2026*) and has been archived with a DOI at Zenodo: https://doi.org/10.5281/zenodo.1552996.

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
