## [Editor Report · eLife Assessment]

In this **important** study, the authors engineered and characterised novel genetically encoded calcium indicators (GECIs) and an analytical tool (CaFire) capable of reporting and quantifying various sub-synaptic events, including miniature synaptic events, with a speed and sensitivity approaching that of intracellular electrophysiological recordings. They present **compelling** data validating this toolset, which will be of interest to neurobiologists studying synaptic calcium dynamics in various model systems.

---

## [Referee Report · Reviewer #1 (Public review)]

Summary:

Chen et al. engineered and characterized a suite of next-generation GECIs for the Drosophila NMJ that allow for the visualization of calcium dynamics within the presynaptic compartment, at presynaptic active zones, and in the postsynaptic compartment. These GECIs include ratiometric presynaptic Scar8m (targeted to synaptic vesicles), ratiometric active zone localized Bar8f (targeted to the scaffold molecule BRP), and postsynaptic SynapGCaMP8m. The authors demonstrate that these new indicators are a large improvement on the widely used GCaMP6 and GCaMP7 series GECIs, with increased speed and sensitivity. They show that presynaptic Scar8m accurately captures presynaptic calcium dynamics with superior sensitivity to the GCaMP6 and GCaMP7 series and with similar kinetics to chemical dyes. The active-zone targeted Bar8f sensor was assessed for the ability to detect release-site specific nanodomain changes, but the authors concluded that this sensor is still too slow to accurately do so. Lastly, the use of postsynaptic SynapGCaMP8m was shown to enable the detection of quantal events with similar resolution to electrophysiological recordings. Finally, the authors developed a Python-based analysis software, CaFire, that enables automated quantification of evoked and spontaneous calcium signals. These tools will greatly expand our ability to detect activity at individual synapses without the need for chemical dyes or electrophysiology.

Strengths:

In this study, the authors rigorously compare their newly engineered GECIs to those previously used at the Drosophila NMJ, highlighting improvements in localization, speed, and sensitivity. These comparisons appropriately substantiate the authors claim that their GECIs are superior to the ones currently in use.

The authors demonstrate the ability of Scar8m to capture subtle changes in presynaptic calcium resulting from differences between MN-Ib and MN-Is terminals and from the induction of presynaptic homeostatic potentiation (PHP), rivaling the sensitivity of chemical dyes.

The improved postsynaptic SynapGCaMP8m is shown to approach the resolution of electrophysiology in resolving quantal events.

The authors created a publicly available pipeline that streamlines and standardizes analysis of calcium imaging data.

---

## [Referee Report · Reviewer #2 (Public review)]

Summary:

Calcium ions play a key role in synaptic transmission and plasticity. To improve calcium measurements at synaptic terminals, previous studies have targeted genetically encoded calcium indicators (GECIs) to pre- and postsynaptic locations. Here, Chen et al. improve these constructs by incorporating the latest GCaMP8 sensors and a stable red fluorescent protein to enable ratiometric measurements. Extensive characterization in the Drosophila neuromuscular junction demonstrates favorable performance of these new constructs relative to previous genetically encoded and synthetic calcium indicators in reporting synaptic calcium events. In addition, they develop a new analysis platform, 'CaFire', to facilitate automated quantification. Impressively, by positioning postsynaptic GCaMP8m near glutamate receptors, the authors show that their sensors can report miniature synaptic events with speed and sensitivity approaching that of intracellular electrophysiological recordings. These new sensors and the analysis platform provide a valuable tool for resolving synaptic events using all-optical methods.

Strength:

The authors present rigorous characterization of their sensors using well-established assays. They employ immunostaining and super-resolution STED microscopy to confirm correct subcellular targeting. Additionally, they quantify response amplitude, rise and decay kinetics, and provide side-by-side comparisons with earlier-generation GECIs and synthetic dyes. Importantly, they show that the new sensors can reproduce known differences in evoked Ca²⁺ responses between distinct nerve terminals. Finally, they present what appears to be the first simultaneous calcium imaging and intracellular mEPSP recording to directly assess the sensitivity of different sensors in detecting individual miniature synaptic events.

The revised version contains extensive new data and clarification that fully addressed my previous concerns. In particular, I appreciate the side-by-side comparison with synthetic calcium indicator OGB-1 and the cytosolic version of GCaMP8m (now presented in Figure 3), which compellingly supports the favorable performance of their new sensors.

Weakness:

I have only one remaining suggestion about the precision of terminology, which I do think is important. The authors clarified in the revision that they "define SNR operationally as the fractional fluorescence change (ΔF/F).", and basically present ΔF/F values whenever they mentioned about SNR. However, if the intention is to present ΔF/F comparisons, I would strongly suggest replacing all mentions of "SNR" in the manuscript with "ΔF/F" or "fractional/relative fluorescence change".

SNR and ΔF/F are fundamentally different quantities, each with a well-defined and distinct meaning: SNR measures fluorescence change relative to baseline fluctuations (noise), whereas ΔF/F measures fluorescence change relative to baseline fluorescence (F₀). While larger ΔF/F values often correlate with improved detectability, SNR also depends on additional factors such as indicator brightness, light collection efficiency, camera noise, and the stability of the experimental preparation. Referring to ΔF/F as SNR can therefore be misleading and may cause confusion for readers, particularly those from quantitative imaging backgrounds. Clarifying the terminology by consistently using ΔF/F would improve conceptual accuracy without requiring any reanalysis of the data.

---

## [Referee Report · Reviewer #3 (Public review)]

Genetically encoded calcium indicators (GECIs) are essential tools in neurobiology and physiology. Technological constraints in targeting and kinetics of previous versions of GECIs have limited their application at the subcellular level. Chen et al. present a set of novel tools that overcome many of these limitations. Through systematic testing in the Drosophila NMJ, they demonstrate improved targeting of GCaMP variants to synaptic compartments and report enhanced brightness and temporal fidelity using members of the GCaMP8 series. These advancements are likely to facilitate more precise investigation of synaptic physiology. This manuscript could be improved by further testing the GECIs across physiologically relevant ranges of activity, including at high frequency and over long imaging sessions. Moreover, while the authors provide a custom software package (CaFire) for Ca2+ imaging analysis, comparisons to existing tools and more guidance for broader usability are needed.

In this revised version, Chen et al. answered most of our concerns. The tools developed here will be useful for the community.

---

## [Author Response]

The following is the authors’ response to the original reviews.

**Reviewer #1**
Chen et al. engineered and characterized a suite of next-generation GECIs for the Drosophila NMJ that allow for the visualization of calcium dynamics within the presynaptic compartment, at presynaptic active zones, and in the postsynaptic compartment. These GECIs include ratiometric presynaptic Scar8m (targeted to synaptic vesicles), ratiometric active zone localized Bar8f (targeted to the scaffold molecule BRP), and postsynaptic SynapGCaMP8m. The authors demonstrate that these new indicators are a large improvement on the widely used GCaMP6 and GCaMP7 series GECIs, with increased speed and sensitivity. They show that presynaptic Scar8m accurately captures presynaptic calcium dynamics with superior sensitivity to the GCaMP6 and GCaMP7 series and with similar kinetics to chemical dyes. The active-zone targeted Bar8f sensor was assessed for the ability to detect release-site-specific nanodomain changes, but the authors concluded that this sensor is still too slow to accurately do so. Lastly, the use of postsynaptic SynapGCaMP8m was shown to enable the detection of quantal events with similar resolution to electrophysiological recordings. Finally, the authors developed a Python-based analysis software, CaFire, that enables automated quantification of evoked and spontaneous calcium signals. These tools will greatly expand our ability to detect activity at individual synapses without the need for chemical dyes or electrophysiology.

We thank this Reviewer for the overall positive assessment of our manuscript and for the incisive comments.

(1) The role of Excel in the pipeline could be more clearly explained. Lines 182-187 could be better worded to indicate that CaFire provides analysis downstream of intensity detection in ImageJ. Moreover, the data type of the exported data, such as .csv or .xlsx, should be indicated instead of 'export to graphical program such as Microsoft Excel'.

We thank the Reviewer for these comments, many of which were shared by the other reviewers. In response, we have now (1) more clearly explained the role of Excel in the CaFire pipeline (lines 677-681), (2) revised the wording in lines 676-679 to indicate that CaFire provides analysis downsteam of intensity detection in ImageJ, and (3) Clarified the exported data type to Excel (lines 677-681). These efforts have improved the clarity and readability of the CaFire analysis pipeline.

(2) In Figure 2A, the 'Excel' step should either be deleted or included as 'data validation' as ImageJ exports don't require MS Excel or any specific software to be analysed. (Also, the graphic used to depict Excel software in Figure 2A is confusing.)

We thank the reviewer for this helpful suggestion. In the Fig. 2A, we have changed the Excel portion and clarified the processing steps in the revised methods. Specifically, we now indicate that ROIs are first selected in Fiji/ImageJ and analyzed to obtain time-series data containing both the time information and the corresponding imaging mean intensity values. These data are then exported to a spreadsheet file (e.g., Excel), which is used to organize the output before being imported into CaFire for subsequent analysis. These changes can be found in the Fig. 2A and methods (lines 676-681).

(3) Figure 2B should include the 'Partition Specification' window (as shown on the GitHub) as well as the threshold selection to give the readers a better understanding of how the tool works.

We absolutely agree with this comment, and have made the suggested changes to the Fig. 2B. In particular, we have replaced the software interface panels and now include windows illustrating the Load File, Peak Detection, and Partition functions. These updated screenshots provide a clearer view of how CaFire is used to load the data, detect events, and perform partition specification for subsequent analysis. We agree these changes will give the readers a better understanding of how the tool works, and we thank the reviewer for this comment.

(4) The presentation of data is well organized throughout the paper. However, in Figure 6C, it is unclear how the heatmaps represent the spatiotemporal fluorescence dynamics of each indicator. Does the signal correspond to a line drawn across the ROI shown in Figure 6B? If so, this should be indicated.

We apologize that the heatmaps were unclear in Fig panel 6C (Fig. 7C in the Current revision). Each heatmap is derived from a one-pixel-wide vertical line within a miniature-event ROI. These heatmaps correspond to the fluorescence change in the indicated SynapGCaMP variant of individual quantal events and their traces shown in Fig. 7C, with a representative image of the baseline and peak fluorescence shown in Fig. 7B. Specifically, we have added the following to the revised Fig. 7C legend:

The corresponding heatmaps below were generated from a single vertical line extracted from a representative miniature-event ROI, and visualize the spatiotemporal fluorescence dynamics (ΔF/F) along that line over time.

(5) In Figure 6D, the addition of non-matched electrophysiology recordings is confusing. Maybe add "at different time points" to the end of the 6D legend, or consider removing the electrophysiology trace from Figure 6D and referring the reader to the traces in Figure 7A for comparison (considering the same point is made more rigorously in Figure 7).

This is a good point, one shared with another reviewer. We apologize this was not clear, and have now revised this part of the figure to remove the electrophysiological traces in what is now Fig. 7 while keeping the paired ones still in what is now Fig. 8A as suggested by the reviewer. We agree this helps to clarify the quantal calcium transients.

(6) In GitHub, an example ImageJ Script for analyzing the images and creating the inputs for CaFire would be helpful to ensure formatting compatibility, especially given potential variability when exporting intensity information for two channels. In the Usage Guide, more information would be helpful, such as how to select ∆R/R, ideally with screenshots of the application being used to analyze example data for both single-channel and two-channel images.

We agree that additional details added to the GitHub would be helpful for users of CaFire. In response, we have now added the following improvements to the GitHub site:

- ImageJ operation screenshots

Step-by-step illustrations of ROI drawing and Multi Measure extraction.

- Example Excel file with time and intensity values

Demonstrates the required data format for CaFire import, including proper headers.

- CaFire loading screenshots for single-channel and dual-channel imaging

Shows how to import GCaMP into Channel 1 and mScarlet into Channel 2.

- Peak Detection and Partition setting screenshots

Visual examples of automatic peak detection, manual correction, and trace partitioning.

- Instructions for ROI Extraction and CaFire Analysis

A written guide describing the full workflow from ROI selection to CaFire data export.

These changes have improved the usability and accessibility of CaFire, and we thank the reviewer for these points.

**Reviewer #2**
Calcium ions play a key role in synaptic transmission and plasticity. To improve calcium measurements at synaptic terminals, previous studies have targeted genetically encoded calcium indicators (GECIs) to pre- and postsynaptic locations. Here, Chen et al. improve these constructs by incorporating the latest GCaMP8 sensors and a stable red fluorescent protein to enable ratiometric measurements. In addition, they develop a new analysis platform, 'CaFire', to facilitate automated quantification. Using these tools, the authors demonstrate favorable properties of their sensors relative to earlier constructs. Impressively, by positioning postsynaptic GCaMP8m near glutamate receptors, they show that their sensors can report miniature synaptic events with speed and sensitivity approaching that of intracellular electrophysiological recordings. These new sensors and the analysis platform provide a valuable tool for resolving synaptic events using all-optical methods.

We thank the Reviewer for their overall positive evaluation and comments.

Major comments:(1) While the authors rigorously compared the response amplitude, rise, and decay kinetics of several sensors, key parameters like brightness and photobleaching rates are not reported. I feel that including this information is important as synaptically tethered sensors, compared to freely diffusible cytosolic indicators, can be especially prone to photobleaching, particularly under the high-intensity illumination and high-magnification conditions required for synaptic imaging. Quantifying baseline brightness and photobleaching rates would add valuable information for researchers intending to adopt these tools, especially in the context of prolonged or high-speed imaging experiments.

This is a good point made by the reviewer, and one we agree will be useful for researchers to be aware. First, it is important to note that the photobleaching and brightness of the sensors will vary depending on the nature of the user’s imaging equipment, which can vary significantly between widefield microscopes (with various LED or halogen light sources for illumination), laser scanning systems (e.g., line scans with confocal systems), or area scanning systems using resonant scanners (as we use in our current study). Under the same imaging settings, GCaMP8f and 8m exhibit comparable baseline fluorescence, whereas GCaMP6f and 6s are noticeably dimmer; because our aim is to assess each reagent’s potential under optimal conditions, we routinely adjust excitation/camera parameters before acquisition to place baseline fluorescence in an appropriate dynamic range. As an important addition to this study, motivated by the reviewer’s comments above, we now directly compare neuronal cytosolic GCaMP8m expression with our Scar8m sensor, showing higher sensitivity with Scar8m (now shown in the new Fig. 3F-H).

Regarding photobleaching, GCaMP signals are generally stable, while mScarlet is more prone to bleaching: in presynaptic area scanned confocal recordings, the mScarlet channel drops by ~15% over 15 secs, whereas GCaMP6s/8f/8m show no obvious bleaching over the same window (lines 549-553). In contrast, presynaptic widefield imaging using an LED system (CCD), GCaMP8f shows ~8% loss over 15 secs (lines 610-611). Similarly, for postsynaptic SynapGCaMP6f/8f/8m, confocal resonant area scans show no obvious bleaching over 60 secs, while widefield shows ~2–5% bleaching over 60 secs (lines 634-638). Finally, in active-zone/BRP calcium imaging (confocal), mScarlet again bleaches by ~15% over 15 s, while GCaMP8f/8m show no obvious bleaching. The mScarlet-channel bleaching can be corrected in Huygens SVI (Bleaching correction or via the Deconvolution Wizard), whereas we avoid applying bleaching correction to the green GCaMP channel when no clear decay is present to prevent introducing artifacts. This information is now added to the methods (lines 548-553).

(2) In several places, the authors compare the performance of their sensors with synthetic calcium dyes, but these comparisons are based on literature values rather than on side-by-side measurements in the same preparation. Given differences in imaging conditions across studies (e.g., illumination, camera sensitivity, and noise), parameters like indicator brightness, SNR, and photobleaching are difficult to compare meaningfully. Additionally, the limited frame rate used in the present study may preclude accurate assessment of rise times relative to fast chemical dyes. These issues weaken the claim made in the abstract that "...a ratiometric presynaptic GCaMP8m sensor accurately captures .. Ca²⁺ changes with superior sensitivity and similar kinetics compared to chemical dyes." The authors should clearly acknowledge these limitations and soften their conclusions. A direct comparison in the same system, if feasible, would greatly strengthen the manuscript.

We absolutely agree with these points made the reviewer, and have made a concerted effort to address them through the following:

We have now directly compared presynaptic calcium responses on the same imaging system using the chemical dye Oregon Green Bapta-1 (OGB-1), one of the primary synthetic calcium indicators used in our field. These experiments reveal that Scar8f exhibits markedly faster kinetics and an improved signal-to-noise ratio compared to OGB-1, with higher peak fluorescence responses (Scar8f: 0.32, OGB-1: 0.23). The rise time constants of the two indicators are comparable (both ~3 msecs), whereas the decay of Scar8f is faster than that of OGB-1 (Scar8f: ~40, OGB-1: ~60), indicating more rapid signal recovery. These results now directly demonstrate the superiority of the new GCaMP8 sensors we have engineered over conventional synthetic dyes, and are now presented in the new Fig. 3A-E of the manuscript.

We agree with the reviewer that, in the original submission, the relatively slow resonant area scans (~115 fps) limited the temporal resolution of our rise time measurements. To address this, we have re-measured the rise time using higher frame-rate line scans (kHz). For Scar8f, the rise time constant was 6.736 msec at ~115 fps resonant area scanned, but shortened to 2.893 msec when imaged at ~303 fps, indicating that the original protocol underestimated the true kinetics. In addition, for Bar8m, area scans at ~118 fps yielded a rise time constant of 9.019 msec, whereas line scans at ~1085 fps reduced the rise time constant to 3.230 msec. These new measurements are now incorporated into the manuscript (Figs. 3,4, and 6) to more accurately reflect the fast kinetics of these indicators.

(3) The authors state that their indicators can now achieve measurements previously attainable with chemical dyes and electrophysiology. I encourage the authors to also consider how their tools might enable new measurements beyond what these traditional techniques allow. For example, while electrophysiology can detect summed mEPSPs across synapses, imaging could go a step further by spatially resolving the synaptic origin of individual mEPSP events. One could, for instance, image MN-Ib and MN-Is simultaneously without silencing either input, and detect mEPSP events specific to each synapse. This would enable synapse-specific mapping of quantal events - something electrophysiology alone cannot provide. Demonstrating even a proof-of-principle along these lines could highlight the unique advantages of the new tools by showing that they not only match previous methods but also enable new types of measurements.

These are excellent points raised by the reviewer. In response, we have done the following:

We have now included a supplemental video as “proof-of-principle” data showing simultaneous imaging of SynapGCaMP8m quantal events at both MN-Is and -Ib, demonstrating that synapse-specific spatial mapping of quantal events can be obtained with this tool (see new Supplemental Video 1).

We have also included an additional discussion of the potential and limitations of these tools for new measurements beyond conventional approaches. This discussion is now presented in lines 419-421 in the manuscript.

(4) For ratiometric measurements, it is important to estimate and subtract background signals in each channel. Without this correction, the computed ratio may be skewed, as background adds an offset to both channels and can distort the ratio. However, it is not clear from the Methods section whether, or how, background fluorescence was measured and subtracted.

This is a good point, and we agree more clarification about how ratiometric measurements were made is needed. In response, we have now added the following to the Methods section (lines 548-568):

Time-lapse videos were stabilized and bleach-corrected prior to analysis, which visibly reduced frame-toframe motion and intensity drift. In the presynaptic and active-zone mScarlet channel, a bleaching factor of ~1.15 was observed during the 15 sec recording. This bleaching can be corrected using the “Bleaching correction” tool in Huygens SVI. For presynaptic and active-zone GCaMP signals, there was minimal bleaching over these short imaging periods. Therefore, the bleaching correction step for GCaMP was skipped. Both GCaMP and mScarlet channels were processed using the default settings in the Huygens SVI “Deconvolution Wizard” (with the exception of the bleaching correction option). Deconvolution was performed using the CMLE algorithm with the Huygens default stopping criterion and a maximum of 30 iterations, such that the algorithm either converged earlier or, if convergence was not reached, was terminated at this 30iteration limit; no other iteration settings were used across the GCaMP series. ROIs were drawn on the processed images using Fiji ImageJ software, and mean fluorescence time courses were extracted for the GCaMP and mScarlet channels, yielding F_GCaMP_(t) and F_mScarlet_(t). F(t)s were imported into CaFire with GCaMP assigned to Channel #1 (signal; required) and mScarlet to Channel #2 (baseline/reference; optional). If desired, the mScarlet signal could be smoothed in CaFire using a user-specified moving-average window to reduce high-frequency noise. In CaFire’s ΔR/R mode, the per-frame ratio was computed as R(t)=F_GCaMP_(t) and F_mScarlet_(t); a baseline ratio R0 was estimated from the pre-stimulus period, and the final response was reported as ΔR/R(t)=[R(t)−R0]/R0, which normalizes GCaMP signals to the co-expressed mScarlet reference and thereby reduces variability arising from differences in sensor expression level or illumination across AZs.

(5) At line 212, the authors claim "... GCaMP8m showing 345.7% higher SNR over GCaMP6s....(Fig. 3D and E) ", yet the cited figure panels do not present any SNR quantification. Figures 3D and E only show response amplitudes and kinetics, which are distinct from SNR. The methods section also does not describe details for how SNR was defined or computed.

This is another good point. We define SNR operationally as the fractional fluorescence change (ΔF/F). Traces were processed with CaFire, which estimates a per-frame baseline F_0_(t) with a user-configurable sliding window and percentile. In the Load File panel, users can specify both the length of the moving baseline window and the desired percentile; the default settings are a 50-point window and the 30th percentile, representing a 101-point window centered on each time point (previous 50 to next 50 samples) and took the lower 30% of values within that window to estimate F_0_(t). The signal was then computed as ΔF/F=[F(t)−F0(t)]/F0(t). This ΔF/F value is what we report as SNR throughout the manuscript and is now discussed explicitly in the revised methods (lines 686-693).

(6) Lines 285-287 "As expected, summed ΔF values scaled strongly and positively with AZ size (Fig. 5F), reflecting a greater number of Cav2 channels at larger AZs". I am not sure about this conclusion. A positive correlation between summed ΔF values and AZ size could simply reflect more GCaMP molecules in larger AZs, which would give rise to larger total fluorescence change even at a given level of calcium increase.

The reviewer makes a good point, one that we agree should be clarified. The reviewer is indeed correct that larger active zones should have more abundant BRP protein, which in turn will lead to a higher abundance of the Bar8f sensor, which should lead to a higher GCaMP response simply by having more of this sensor. However, the inclusion of the ratiometric mScarlet protein should normalize the response accurately, correcting for this confound, in which the higher abundance of GCaMP should be offset (normalized) by the equally (stoichiometric) higher abundance of mScarlet. Therefore, when the ∆R/R is calculated, the differences in GCaMP abundance at each AZ should be corrected for the ratiometric analysis. We now use an improved BRP::mScarlet3::GCaMP8m (Bar8m) and compute ΔR/R with R(t)=F_GCaMP8m_/F_mScarlet3_. ROIs were drawn over individual AZs (Fig. 6B). CaFire estimated R0 with a sliding 101-point window using the lowest 10% of values, and responses were reported as ΔR/R=[R−R0]/R0. Area-scan examples (118 fps) show robust ΔR/R transients (peaks ≈1.90 and 3.28; tau rise ≈9.0–9.3 ms; Fig. 6C, middle).

We have now made these points more clearly in the manuscript (lines 700-704) and moved the Bar8f intensity vs active zone size data to Table S1. Together, these revisions improve the indicator-abundance confound (via mScarlet normalization).

(6) Lines 313-314: "SynapGCaMP quantal signals appeared to qualitatively reflect the same events measured with electrophysiological recordings (Fig. 6D)." This statement is quite confusing. In Figure 6D, the corresponding calcium and ephys traces look completely different and appear to reflect distinct sets of events. It was only after reading Figure 7 that I realized the traces shown in Figure 6D might not have been recorded simultaneously. The authors should clarify this point.

Yes, we absolutely agree with this point, one shared by Reviewer 1. In response, we have removed the electrophysiological traces in Fig. 6 to clarify that just the calcium responses are shown, and save the direct comparison for the Fig. 7 data (now revised Fig. 8).

(8) Lines 310-313: "SynapGCaMP8m .... striking an optimal balance between speed and sensitivity", and Lines 314-316: "We conclude that SynapGCaMP8m is an optimal indicator to measure quantal transmission events at the synapse." Statements like these are subjective. In the authors' own comparison, GCaMP8m is significantly slower than GCaMP8f (at least in terms of decay time), despite having a moderately higher response amplitude. It is therefore unclear why GCaMP8m is considered 'optimal'. The authors should clarify this point or explain their rationale for prioritizing response amplitude over speed in the context of their application.

This is another good point that we agree with, as the “optimal” sensor will of course depend on the user’s objectives. Hence, we used the term “an optimal sensor” to indicate it is what we believed to be the best one for our own uses. However, this point should be clarified and better discussed. In response, we have revised the relevant sections of the manuscript to better define why we chose the 8m sensors to strike an optimal balance of speed and sensitivity for our uses, and go on to discuss situations in which other sensor variants might be better suited. These are now presented in lines 223-236 in the revised manuscript, and we thank the reviewer for making these comments, which have improved our study.

Minor comments(1) Please include the following information in the Methods section:(a) For Figures 3 and 4, specify how action potentials were evoked. What type of electrodes were used, where were they placed, and what amount of current or voltage was applied?

We apologize for neglecting to include this information in the original submission. We have now added this information to the revised Methods section (lines 537-543).

(b) For imaging experiments, provide information on the filter sets used for each imaging channel, and describe how acquisition was alternated or synchronized between the green and red channels in ratiometric measurements. Additionally, please report the typical illumination intensity (in mW/mm²) for each experimental condition.

We thank the reviewer for this helpful comment. We have now added detailed information about the imaging configuration to the Methods (lines 512-528) with the following:

Ca2+ imaging was conducted using a Nikon A1R resonant scanning confocal microscope equipped with a 60x/1.0 NA water-immersion objective (refractive index 1.33). GCaMP signals were acquired using the FITC/GFP channel (488-nm laser excitation; emission collected with a 525/50-nm band-pass filter), and mScarlet/mCherry signals were acquired using the TRITC/mCherry channel (561-nm laser excitation; emission collected with a 595/50-nm band-pass filter). ROIs focused on terminal boutons of MN-Ib or -Is motor neurons. For both channels, the confocal pinhole was set to a fixed diameter of 117.5 µm (approximately three Airy units under these conditions), which increases signal collection while maintaining adequate optical sectioning. Images were acquired as 256 × 64 pixel frames (two 12-bit channels) using bidirectional resonant scanning at a frame rate of ~118 frames/s; the scan zoom in NIS-Elements was adjusted so that this field of view encompassed the entire neuromuscular junction and was kept constant across experiments. In ratiometric recordings, the 488-nm (GCaMP) and 561-nm (mScarlet) channels were acquired in a sequential dual-channel mode using the same bidirectional resonant scan settings: for each time point, a frame was first collected in the green channel and then immediately in the red channel, introducing a small, fixed frame-to-frame temporal offset while preserving matched spatial sampling of the two channels.

Directly measuring the absolute laser power at the specimen plane (and thus reporting illumination intensity in mW/mm²) is technically challenging on this resonant-scanning system, because it would require inserting a power sensor into the beam path and perturbing the optical alignment; consequently, we are unable to provide reliable absolute mW/mm² values. Instead, we now report all relevant acquisition parameters (objective, numerical aperture, refractive index, pinhole size, scan format, frame rate, and fixed laser/detector settings) and note that laser powers were kept constant within each experimental series and chosen to minimize bleaching and phototoxicity while maintaining an adequate signal-to-noise ratio. We have now added the details requested in the revised Methods section (lines 512-535), including information about the filter sets, acquisition settings, and typical illumination intensity.

(2) Please clarify what the thin versus thick traces represent in Figures 3D, 3F, 4C, and 4E. Are the thin traces individual trials from the same experiment, or from different experiments/animals? Does the thick trace represent the mean/median across those trials, a fitted curve, or a representative example?

We apologize this was not more clear in the original submission. Thin traces are individual stimulus-evoked trials (“sweeps”) acquired sequentially from the same muscle/NMJ in a single preparation; the panel is shown as a representative example of recordings collected across animals. The thick colored trace is the trialaveraged waveform (arithmetic mean) of those thin traces after alignment to stimulus onset and baseline subtraction (no additional smoothing beyond what is stated in Methods). The thick black curve over the decay phase is a single-exponential fit used to estimate τ. Specifically, we fit the decay segment by linear regression on the natural-log–transformed baseline-subtracted signal, which is equivalent to fitting *y* = *ypeak·e−t/τdecay* over the decay window (revised Fig.4D and Fig.5C legends).

(3) Please clarify what the reported sample size (n) represents. Does it indicate the number of experimental repeats, the number of boutons or PSDs, or the number of animals?

Again, we apologize this was not clear. (n) refers to the number of animals (biological replicates), which is reported in Supplementary Table 1. All imaging was performed at muscle 6, abdominal segment A3. Per preparation, we imaged 1-2 NMJs in total, with each imaging targeting 2–3 terminal boutons at the target NMJ and acquired 2–3 imaging stacks choosing different terminal boutons per NMJ. For the standard stimulation protocol, we delivered 1 Hz stimulation for 1ms and captured 14 stimuli in a 15s time series imaging (lines 730-736).

**Reviewer #3**
Genetically encoded calcium indicators (GECIs) are essential tools in neurobiology and physiology. Technological constraints in targeting and kinetics of previous versions of GECIs have limited their application at the subcellular level. Chen et al. present a set of novel tools that overcome many of these limitations. Through systematic testing in the Drosophila NMJ, they demonstrate improved targeting of GCaMP variants to synaptic compartments and report enhanced brightness and temporal fidelity using members of the GCaMP8 series. These advancements are likely to facilitate more precise investigation of synaptic physiology.This is a comprehensive and detailed manuscript that introduces and validates new GECI tools optimized for the study of neurotransmission and neuronal excitability. These tools are likely to be highly impactful across neuroscience subfields. The authors are commended for publicly sharing their imaging software.This manuscript could be improved by further testing the GECIs across physiologically relevant ranges of activity, including at high frequency and over long imaging sessions. The authors provide a custom software package (CaFire) for Ca2+ imaging analysis; however, to improve clarity and utility for future users, we recommend providing references to existing Ca2+ imaging tools for context and elaborating on some conceptual and methodological aspects, with more guidance for broader usability. These enhancements would strengthen this already strong manuscript.

We thank the Reviewer for their overall positive evaluation and comments.

Major comments:(1) Evaluation of the performance of new GECI variants using physiologically relevant stimuli and frequency. The authors took initial steps towards this goal, but it would be helpful to determine the performance of the different GECIs at higher electrical stimulation frequencies (at least as high as 20 Hz) and for longer (10 seconds) (Newman et al, 2017). This will help scientists choose the right GECI for studies testing the reliability of synaptic transmission, which generally requires prolonged highfrequency stimulation.

We appreciate this point by the reviewer and agree it would be of interest to evaluate sensor performance with higher frequency stimulation and for a longer duration. In response, we performed a variety of stimulation protocols at high intensities and times, but found the data to be difficult to separate individual responses given the decay kinetics of all calcium sensors. Hence, we elected not to include these in the revised manuscript. However, we have now included an evaluation of the sensors with 20 Hz electrical stimulation for ~1 sec using a direct comparison of Scar8f with OGB-1. These data are now presented in a new Fig. 3D,E and discussed in the manuscript (lines 396-403).

(2) CaFire.The authors mention, in line 182: 'Current approaches to analyze synaptic Ca2+ imaging data either repurpose software designed to analyze electrophysiological data or use custom software developed by groups for their own specific needs.' References should be provided. CaImAn comes to mind (Giovannucci et al., 2019, eLife), but we think there are other software programs aimed at analyzing Ca2+ imaging data that would permit such analysis.

Thank you for the thoughtful question. At this stage, we’re unable to provide a direct comparison with existing analysis workflows. In surveying prior studies that analyze Drosophila NMJ Ca²⁺ imaging traces, we found that most groups preprocess images in Fiji/ImageJ and then rely on their own custom-made MATLAB or Python scripts for downstream analysis (see Blum et al. 2021; Xing and Wu 2018). Because these pipelines vary widely across labs, a standardized head-to-head evaluation isn’t currently feasible. With CaFire, our goal is to offer a simple, accessible tool that does not require coding experience and minimizes variability introduced by custom scripts. We designed CaFire to lower the barrier to entry, promote reproducibility, and make quantal event analysis more consistent across users. We have added references to the sentence mentioned above.

Regarding existing software that the reviewer mentioned – CaImAn (Giovannucci et al. 2019): We evaluated CaImAn, which is a powerful framework designed for large-scale, multicellular calcium imaging (e.g., motion correction, denoising, and automated cell/ROI extraction). However, it is not optimized for the per-event kinetics central to our project - such as extracting rise and decay times for individual quantal events at single synapses. Achieving this level of granularity would typically require additional custom Python scripting and parameter tuning within CaImAn’s code-centric interface. This runs counter to CaFire’s design goals of a nocode, task-focused workflow that enables users to analyze miniature events quickly and consistently without specialized programming expertise.

Regarding Igor Pro (WaveMetrics), (Müller et al. 2012): Igor Pro is another platform that can be used to analyze calcium imaging signals. However, it is commercial (paid) software and generally requires substantial custom scripting to fit the specific analyses we need. In practice, it does not offer a simple, open-source, point-and-click path to per-event kinetic quantification, which is what CaFire is designed to provide.

The authors should be commended for making their software publicly available, but there are some questions:How does CaFire compare to existing tools?

As mentioned above, we have not been able to adapt the custom scripts used by various labs for our purposes, including software developed in MatLab (Blum et al. 2021), Python (Xing and Wu 2018), and Igor (Müller et al. 2012). Some in the field do use semi-publically available software, including Nikon Elements (Chen and Huang 2017) and CaImAn (Giovannucci et al. 2019). However, these platforms are not optimized for the per-event kinetics central to our project - such as extracting rise and decay times for individual quantal events at single synapses. We have added more details about CaFire, mainly focusing on the workflow and measurements, highlighting the superiority of CaFire, showing that CaFire provides a no-code, standardized pipeline with automated miniature-event detection and per-event metrics (e.g., amplitude, rise time τ, decay time τ), optional ΔR/R support, and auto-partition feature. Collectively, these features make CaFire simpler to operate without programming expertise, more transparent and reproducible across users, and better aligned with the event-level kinetics required for this project.

Very few details about the Huygens deconvolution algorithms and input settings were provided in the methods or text (outside of MLE algorithm used in STED images, which was not Ca2+ imaging). Was it blind deconvolution? Did the team distill the point-spread function for the fluorophores? Were both channels processed for ratiometric imaging? Were the same settings used for each channel? Importantly, please include SVI Huygens in the 'Software and Algorithms' Section of the methods.

We thank the reviewer for raising this important point. We have now expanded the Methods to describe our use of Huygens in more detail and have added SVI Huygens Professional (Scientific Volume Imaging, Hilversum, The Netherlands) to the “Software and Algorithms” section. For Ca²⁺ imaging data, time-lapse stacks were processed in the Huygens Deconvolution Wizard using the standard estimation algorithm (CMLE). This is not a blind deconvolution procedure. Instead, Huygens computes a theoretical point-spread function (PSF) from the full acquisition metadata (objective NA, refractive index, voxel size/sampling, pinhole, excitation/emission wavelengths, etc.); if refractive index values are provided and there is a mismatch, the PSF is adjusted to account for spherical aberration. We did not experimentally distill PSFs from bead measurements, as Huygens’ theoretical PSFs are sufficient for our data.

Both green (GCaMP) and red (mScarlet) channels were processed for ratiometric imaging using the same workflow (stabilization, optional bleaching correction, and deconvolution within Huygens). For each channel, the PSF, background, and SNR were estimated automatically by the same built-in algorithms, so the underlying procedures were identical even though the numerical values differ between channels because of their distinct wavelengths and noise characteristics. Importantly, Huygens normalizes each PSF to unit total intensity, such that the deconvolution itself does not add or remove signal and therefore preserves intensity ratios between channels; only background subtraction and bleaching correction can change absolute fluorescence values. For the mScarlet channel, where we observed modest bleaching (~1.10 over 15 sec), we applied Huygens’ bleaching correction and visually verified that similar structures maintained comparable intensities after correction. For presynaptic GCaMP signals, bleaching over these short recordings was negligible, so we omitted the bleaching-correction step to avoid introducing multiplicative artifacts. This workflow ensures that ratiometric ΔR/R measurements are based on consistently processed, intensity-conserving deconvolved images in both channels.

The number of deconvolution iterations could have had an effect when comparing GCAMP series; please provide an average number of iterations used for at least one experiment. For example, Figure 3, Syt::GCAMP6s, Scar8f & Scar8m, and, if applicable, the maximum number of permissible iterations.

We thank the reviewer for this comment. For all Ca²⁺ imaging datasets, deconvolution in Huygens was performed using the recommended default settings of the CMLE algorithm with a maximum of 30 iterations. The stopping criterion was left at the Huygens default, so the algorithm either converged earlier or, if convergence was not reached, terminated at this 30-iteration limit. No other iteration settings were used across the GCaMP series (lines 555-559).

Please clarify if the 'Express' settings in Huygens changed algorithms or shifted input parameters.

We appreciate the reviewer’s question regarding the Huygens “Express” settings. For clarity, we note that all Ca²⁺ imaging data reported in this manuscript were deconvolved using the “Deconvolution Wizard”, not the “Deconvolution Express” mode. In the Wizard, we explicitly selected the CMLE algorithm (or GMLE in a few STED-related cases as recommended by SVI), using the recommended maximum of 30 iterations, and other recommended settings while allowing Huygens to auto-estimate background and SNR for each channel.Bleaching correction was toggled manually per channel (applied to mScarlet when bleaching was evident, omitted for GCaMP when bleaching was negligible), as described in the revised Methods (lines 553-559).

By contrast, the Deconvolution Express tool in Huygens is a fully automated front-end that can internally adjust both the choice of deconvolution algorithm (e.g., CMLE vs. GMLE/QMLE) and key input parameters such as SNR, number of iterations, and quality threshold based on the selected “smart profile” and the image metadata. In preliminary tests on our datasets, Express sometimes produced results that were either overly smoothed or showed subtle artifacts, so we did not use it for any data included in this study. Instead, we relied exclusively on the Wizard with explicitly controlled settings to ensure consistency and transparency across all GCaMP series and ratiometric analyses.

We suggest including a sample data set, perhaps in Excel, so that future users can beta test on and organize their data in a similar fashion.

We agree that this would be useful, a point shared by R1 above. In response, we have added a sample data set to the GitHub site and included sample ImageJ data along with screenshots to explain the analysis in more detail. These improvements are discussed in the manuscript (lines 705-708).

(3) While the challenges of AZ imaging are mentioned, it is not discussed how the authors tackled each one. What is defined as an active zone? Active zones are usually identified under electron microscopy. Arguably, the limitation of GCaMP-based sensors targeted to individual AZs, being unable to resolve local Ca2+ changes at individual boutons reliably, might be incorrect. This could be a limitation of the optical setup being used here. Please discuss further. What sensor performance do we need to achieve this performance level, and/or what optical setup would we need to resolve such signals?

We appreciate the reviewer’s thoughtful comments and agree that the technical challenges of active zone (AZ) Ca²⁺ imaging merit further clarification. We defined AZs, as is the convention in our field, as individual BRP puncta at NMJs. These BRP puncta co-colocalize with individual puncta of other AZ components, including CAC, RBP, Unc13, etc. ROIs were drawn tightly over individual BRP puncta and only clearly separable spots were included.

To tackle the specific obstacles of AZ imaging (small signal volume, high AZ density, and limited photon budget at high frame rates), we implemented both improved sensors and optimized analysis (Fig. 6). First, we introduced a ratiometric AZ-targeted indicator, BRP::mScarlet3::GCaMP8m (Bar8m), and computed ΔR/R with ΔR/R with R(t)=F_GCaMP8m_/F_mScarlet3_. ROIs were drawn over individual AZs (Fig. 6B). Under our standard resonant area-scan conditions (~118 fps), Bar8m produces robust ΔR/R transients at individual AZs (example peaks ≈ 3.28; τ_rise_≈9.0 ms; Fig. 6C, middle), indicating that single-AZ signals can be detected reproducibly when AZs are optically resolvable.

Second, we increased temporal resolution using high-speed Galvano line-scan imaging (~1058 fps), which markedly sharpened the apparent kinetics (τ_rise_≈3.23 ms) and revealed greater between-AZ variability (Fig. 6C, right; 6D–E). Population analyses show that line scans yield much faster rise times than area scans (Fig. 6D) and a dramatically higher fraction of significantly different AZ pairs (8.28% and 4.14% in 8f and 8m areascan vs 78.62% in 8m line-scan, lines 721-725), uncovering pronounced AZ-to-AZ heterogeneity in Ca²⁺ signals. Together, these revisions demonstrate that under our current confocal configuration, AZ-targeted GCaMP8m can indeed resolve local Ca²⁺ changes at individual, optically isolated boutons.

We have revised the Discussion to clarify that our original statement about the limitations of AZ-targeted GCaMPs refers specifically to this combination of sensor and optical setup, rather than an absolute limitation of AZ-level Ca²⁺ imaging. In our view, further improvements in baseline brightness and dynamic range (ΔF/F or ΔR/R per action potential), combined with sub-millisecond kinetics and minimal buffering, together with optical configurations that provide smaller effective PSFs and higher photon collection (e.g., higher-NA objectives, optimized 2-photon or fast line-scan modalities, and potentially super-resolution approaches applied to AZ-localized indicators), are likely to be required to achieve routine, high-fidelity Ca²⁺ measurements at every individual AZ within a neuromuscular junction.

(4) In Figure 5: Only GCAMP8f (Bar8f fusion protein) is tested here. Consider including testing with GCAMP8m. This is particularly relevant given that GCAMP8m was a more successful GECI for subcellular post-synaptic imaging in Figure 6.

We appreciate this point and request by Reviewer 3. The main limitation for detecting local calcium changes at AZs is the speed of the calcium sensor, and hence we used the fastest available (GCaMP8f) to test the Bar8f sensor. While replacing GCaMP8f with GCaMP8m would indeed be predicted to enhance sensitivity (SNR), since GCaMP8m does not have faster kinetics relative to GCaMP8f, it is unlikely to be a more successful GECI for visualizing local calcium differences at AZs.

That being said, we agree that the Bar8m tool, including the improved mScarlet3 indicator, would likely be of interest and use to the field. Fortunately, we had engineered the Bar8m sensor while this manuscript was in review, and just recently received transgenic flies. We have evaluated this sensor, as requested by the reviewer, and included our findings in Fig. 1 and 6. In short, while the sensitivity is indeed enhanced in Bar8m compared to Bar8f, the kinetics remain insufficient to capture local AZ signals. These findings are discussed in the revised manuscript (lines 424-442, 719-730), and we appreciate the reviewer for raising these important points.

In earlier experiments, Bar8f yielded relatively weak fluorescence, so we traded frame rate for image quality during resonant area scans (~60 fps). After switching to Bar8m, the signal was bright enough to restore our standard 118 fps area-scan setting. Nevertheless, even with dual-channel resonant area scans and ratiometric (GCaMP/mScarlet) analysis, AZ-to-AZ heterogeneity remained difficult to resolve. Because Ca²⁺ influx at individual active zones evolves on sub-millisecond timescales, we adopted a high-speed singlechannel Galvano line-scan (~1 kHz) to capture these rapid transients. We first acquired a brief area image to localize AZ puncta, then positioned the line-scan ROI through the center of the selected AZ. This configuration provided the temporal resolution needed to uncover heterogeneity that was under-sampled in area-scan data. Consistent with this, Bar8m line-scan data showed markedly higher AZ heterogeneity (significant AZ-pair rate ~79%, vs. ~8% for Bar8f area scans and ~4% for Bar8m area scans), highlighting Bar8m’s suitability for quantifying AZ diversity. We have updated the text, Methods, and figure legend accordingly (tell reviewer where to find everything).

(5) Figure 5D and associated datasets: Why was Interquartile Range (IQR) testing used instead of ZScoring? Generally, IQR is used when the data is heavily skewed or is not normally distributed. Normality was tested using the D'Agostino & Pearson omnibus normality test and found that normality was not violated. Please explain your reasoning for the approach in statistical testing. Correlation coefficients in Figures 5 E & F should also be reported on the graph, not just the table. In Supplementary Table 1. The sub-table between 4D-F and 5E-F, which describes the IQR, should be labeled as such and contain identifiers in the rows describing which quartile is described. The table description should be below. We would recommend a brief table description for each sub-table.

Thank you for this helpful suggestion. We have updated the analysis in two complementary ways. First, we now perform paired two-tailed t-tests between every two AZs within the same preparation (pairwise AZ–AZ comparisons of peak responses). At α<0.05, the fraction of significant AZ pairs is ~79% for Bar8m line-scan data versus ~8% for Bar8f area-scan data, indicating markedly greater AZ-to-AZ diversity when measured at high temporal resolution. Second, for visually marking the outlying AZs, we re-computed the IQR (Q1–Q3) based on the individual values collected from each AZs(15 data points per AZ, 30 AZs for each genotype), and marked AZs whose mean response falls above Q3 or below Q1; IQR is used here solely as a robust dispersion reference rather than for hypothesis testing. Both analyses support the same observation: Bar8m line-scan data reveal substantially higher AZ heterogeneity than Bar8f and Bar8m area-scan data. We have revised the Methods, figure panels, and legends accordingly (t-test details; explicit “IQR (Q1–Q3)” labeling; significant AZ-pair rates reported on the plots) (lines 719-730).

(6) Figure 6 and associated data. The authors mention: ' SynapGCaMP quantal signals appeared to qualitatively reflect the same events measured with electrophysiological recordings (Fig. 6D).' If that was the case, shouldn't the ephys and optical signal show some sort of correlation? The data presented in Figure 6D show no such correlation. Where do these signals come from? It is important to show the ROIs on a reference image.

We apologize this was not clear, as similar points were raised by R1 and R2. We were just showing separate (uncorrelated) sample traces of electrophysiological and calcium imaging data. Given how confusing this presentation turned out to be, and the fact that we show the correlated ephys and calcium imaging events in Fig. 7, we have elected to remove the uncorrelated electrophysiological events in Fig. 6 to just focus on the calcium imaging events (now Figures 7 and 8).

Figure 7B: Were Ca2+ transients not associated with mEPSPs ever detected? What is the rate of such events?

This is an astute question. Yes indeed, during simultaneous calcium imaging and current clamp electrophysiology recordings, we occasionally observed GCaMP transients without a detectable mEPSP in the electrophysiological trace. This may reflect the detection limit of electrophysiology for very small minis; with our noise level and the technical limitation of the recording rig, events < ~0.2 mV cannot be reliably detected, whereas the optical signal from the same quantal event might still be detected. The fraction of calcium-only events was ~1–10% of all optical miniature events, depending on genotype (higher in lines with smaller average minis). These calcium-only detections were low-amplitude and clustered near the optical threshold (lines 361-365).

Minor comments(1) It should be mentioned in the text or figure legend whether images in Figure 1 were deconvolved, particularly since image pre-processing is only discussed in Figure 2 and after.

We thank the reviewer for pointing this out. Yes, the confocal images shown in Figure 1 were also deconvolved in Huygens using the CMLE-based workflow described in the revised Methods. We applied deconvolution to improve contrast, reduce out-of-focus blur, and better resolve the morphology of presynaptic boutons, active zones, and postsynaptic structures, so that the localization of each sensor is more clearly visualized. We have now explicitly stated in the Fig. 1 legend and Methods (lines 575-577) that these images were deconvolved prior to display.

(2) The abbreviation, SNR, signal-to-noise ratio, is not defined in the text.

We have corrected this error and thank the reviewer for pointing this out.

(3) Please comment on the availability of fly stocks and molecular constructs.

We have clarified that all fly stocks and molecular constructs will be shared upon request (lines 747-750). We are also in the process of depositing the new Scar8f/m, Bar8f/m, and SynapGCaMP sensors to the Bloomington Drosophila Stock Center for public dissemination.

(4) Please add detection wavelengths and filter cube information for live imaging experiments for both confocal and widefield.

We thank the reviewer for this helpful suggestion. We have now added the detection wavelengths and filter cube configurations for both confocal and widefield live imaging to the Methods.

For confocal imaging, GCaMP signals were acquired on a Nikon A1R system using the FITC/GFP channel (488-nm laser excitation; emission collected with a 525/50-nm band-pass filter), and mScarlet signals were acquired using the TRITC/mCherry channel (561-nm laser excitation; emission collected with a 595/50-nm band-pass filter). Both channels were detected with GaAsP detectors under the same pinhole and scan settings described above (lines 512-517).

For widefield imaging, GCaMP was recorded using a GFP filter cube (LED excitation ~470/40 nm; emission ~525/50 nm), which is now explicitly described in the revised Methods section (lines 632-633).

(5) Please include a mini frequency analysis in Supplemental Figure S1.

We apologize for not including this information in the original submission. This is now included in the Supplemental Figure S1.

(6) In Figure S1B, consider flipping the order of EPSP (currently middle) and mEPSP (currently left), to easily guide the reader through the quantification of Figure S1A (EPSPs, top traces & mEPSPs, bottom traces).

We agree these modifications would improve readability and clarity. We have now re-ordered the electrophysiological quantifications in Fig. S1B as requested by the reviewer.

(7) Figure 6C: Consider labeling with sensor name instead of GFP.

We agree here as well, and have removed “GFP” and instead added the GCaMP variant to the heatmap in Fig. 7C.

(8) Figure 6E, 7B, 7E: Main statistical differences highlighting sensor performance should be represented on the figures for clarity.

We did not show these differences in the original submission in an effort to keep the figures “clean” and for clarity, putting the detailed statistical significance in Table S1. However, we agree with the reviewer that it would be easier to see these in the Fig. 6E and 7B,E graphs. This information has now been added the Figs. 7 and 8.

(9) Please report if the significance tested between the ephys mini (WT vs IIB-/-, WT vs IIA-/-, IIB-/- vs IIA-/-) is the same as for Ca2+ mini (WT vs IIB-/-, WT vs IIA-/-, IIB-/- vs IIA-/-). These should also exhibit a very high correlation (mEPSP (mV) vs Ca2+ mini deltaF/F). These tests would significantly strengthen the final statement of "SynapGCaMP8m can capture physiologically relevant differences in quantal events with similar sensitivity as electrophysiology."

We agree that adding the more detailed statistical analysis requested by the reviewer would strengthen the evidence for the resolution of quantal calcium imaging using SynapGCaMP8m. We have included the statistical significance between the ephys and calcium minis in Fig. 8 and included the following in the revised methods (lines 358-361), the Fig. 8 legend and Table S1:

Using two-sample Kolmogorov–Smirnov (K–S) tests, we found that SynapGCaMP8m Ca²⁺ minis (ΔF/F, Fig. 8E) differ significantly across all genotype pairs (WT vs IIB^-/-^, WT vs IIA^-/-^, IIB^-/-^ vs IIA^-/-^; all p < 0.0001). The genotype rank order of the group means (± SEM) is IIB^-/-^ > WT > IIA^-/-^ (0.967 ± 0.036; 0.713 ± 0.021; 0.427 ± 0.017; n=69, 65, 59). For electrophysiological minis (mEPSP amplitude, Fig. 8F), K–S tests likewise show significant differences for the same comparisons (all p < 0.0001) with D statistics of 0.1854, 0.3647, and 0.4043 (WT vs IIB^-/-^, WT vs IIA^-/-^, IIB^-/-^ vs IIA^-/-^, respectively). Group means (± SEM) again follow IIB^-/-^ > WT > IIA^-/-^ (0.824 ± 0.017 mV; 0.636 ± 0.015 mV; 0.383 ± 0.007 mV; n=41 each). These K–S results demonstrate identical significance and rank order across modalities, supporting our conclusion that SynapGCaMP8m resolves physiologically relevant quantal differences with sensitivity comparable to electrophysiology.

References

Blum, Ian D., Mehmet F. Keleş, El-Sayed Baz, Emily Han, Kristen Park, Skylar Luu, Habon Issa, Matt Brown, Margaret C. W. Ho, Masashi Tabuchi, Sha Liu, and Mark N. Wu. 2021. 'Astroglial Calcium Signaling Encodes Sleep Need in Drosophila', Current Biology, 31: 150-62.e7.

Chen, Y., and L. M. Huang. 2017. 'A simple and fast method to image calcium activity of neurons from intact dorsal root ganglia using fluorescent chemical Ca(2+) indicators', Mol Pain, 13: 1744806917748051.

Giovannucci, Andrea, Johannes Friedrich, Pat Gunn, Jérémie Kalfon, Brandon L. Brown, Sue Ann Koay, Jiannis Taxidis, Farzaneh Najafi, Jeffrey L. Gauthier, Pengcheng Zhou, Baljit S. Khakh, David W. Tank, Dmitri B. Chklovskii, and Eftychios A. Pnevmatikakis. 2019. 'CaImAn an open source tool for scalable calcium imaging data analysis', eLife, 8: e38173.

Müller, M., K. S. Liu, S. J. Sigrist, and G. W. Davis. 2012. 'RIM controls homeostatic plasticity through modulation of the readily-releasable vesicle pool', J Neurosci, 32: 16574-85.

Wu, Yifan, Keimpe Wierda, Katlijn Vints, Yu-Chun Huang, Valerie Uytterhoeven, Sahil Loomba, Fran Laenen, Marieke Hoekstra, Miranda C. Dyson, Sheng Huang, Chengji Piao, Jiawen Chen, Sambashiva Banala, Chien-Chun Chen, El-Sayed Baz, Luke Lavis, Dion Dickman, Natalia V. Gounko, Stephan Sigrist, Patrik Verstreken, and Sha Liu. 2025. 'Presynaptic Release Probability Determines the Need for Sleep', bioRxiv: 2025.10.16.682770.

Xing, Xiaomin, and Chun-Fang Wu. 2018. 'Unraveling Synaptic GCaMP Signals: Differential Excitability and Clearance Mechanisms Underlying Distinct Ca^2+^ Dynamics in Tonic and Phasic Excitatory, and Aminergic Modulatory Motor Terminals in *Drosophila*', eneuro, 5: ENEURO.0362-17.2018.